# Multi-species host range of staphylococcal phages isolated from wastewater

Pauline C. Göller[1], Tabea Elsener[1], Dominic Lorgé[1], Natasa Radulovic[1], Viona Bernardi[1], Annika Naumann[1], Nesrine Amri[1], Ekaterina Khatchatourova[1], Felipe Hernandes Coutinho[2], Martin J. Loessner [1] & Elena Gómez-Sanz [1,3]✉

The host range of bacteriophages defines their impact on bacterial communities and genome diversity. Here, we characterize 94 novel staphylococcal phages from wastewater and establish their host range on a diversified panel of 117 staphylococci from 29 species. Using this high-resolution phage-bacteria interaction matrix, we unveil a multi-species host range as a dominant trait of the isolated staphylococcal phages. Phage genome sequencing shows this pattern to prevail irrespective of taxonomy. Network analysis between phage-infected bacteria reveals that hosts from multiple species, ecosystems, and drug-resistance pheno-types share numerous phages. Lastly, we show that phages throughout this network can package foreign genetic material enclosing an antibiotic resistance marker at various frequencies. Our findings indicate a weak host specialism of the tested phages, and therefore their potential to promote horizontal gene transfer in this environment.

[1] Institute of Food, Nutrition and Health, ETH Zurich, 8092 Zurich, Switzerland. [2] Departamento de Producción Vegetal y Microbiología, Universidad Miguel Hernández, San Juan de Alicante, Spain. [3] Área de Microbiología Molecular, Centro de Investigación Biomédica de La Rioja (CIBIR), Logroño, Spain. ✉email: elena.gomez@hest.ethz.ch

Bacteriophages (phages) are the most abundant biological entities on Earth, and yet the understanding of the association between bacteria and their infecting phages remains limited. The phage host range, defined as the taxonomic breadth of bacteria a phage can successfully infect (reviewed in[1]), is a central trait to understand in phage biology (reviewed in[2]). Labor-intensive infection assays showed that host ranges diversify from narrow to broad[3], while "broad" and "narrow" are partially conditioned by the genetic diversity of challenged hosts[2]. To date, most isolated phages (>85%) belong to the order *Caudovirales*[4] and are reported as specialists with narrow host ranges. However, a few network studies from a compilation of published data or marine viruses find that phages can infect a multitude of hosts and that different phage types predate each bacterial species[5–7]. Such interactions have profound implications on how phages influence bacterial community composition and ecology[8,9], or facilitate horizontal gene transfer (HGT)[10–12].

Bacteria of the genus *Staphylococcus* are part of the natural skin microbiota of mammals but also represent life-threatening pathogens due to their increasing virulence and antibiotic resistance repertoire. Based on their ability to produce coagulase, staphylococci are divided into the traditionally more pathogenic coagulase-positive staphylococci (CoPS), with *S. aureus* as the major species, and coagulase-negative staphylococci (CoNS), such as *S. epidermidis*. Dissociated from clinical manifestations and based on multilocus data, a refined phylogeny for staphylococcal species into 15 species cluster and six species groups was recently suggested[13]. Today, CoNS are increasingly recognized as major nosocomial pathogens with limited treatment options due to a large proportion of antibiotic-resistant strains[14]. They are regarded as important reservoirs of antimicrobial resistance determinants that could spread to clinical most critical species. Transduction, the horizontal transfer of genetic material by phages, is thereby perceived as the primary route for genetic exchange (reviewed in[15]). This notion is supported by the exceptional high-frequency of prophages are detected in the genomes of sequenced staphylococci[15,16]. To date, all published staphylococcal phages exhibit a myovirus, siphovirus, or podovirus morphology and belong to the taxonomic families of *Herelleviridae*, *Siphoviridae*, and *Podoviridae*, respectively. According to their shared gene content, staphylococcal phages are additionally grouped into four phylogenetic clusters (A-D) and one singleton (phage SPbeta-like). All phages within a cluster share discrete genome lengths. Here, the temperate siphoviruses with genome sizes ~40 kb make up cluster B, and presumably virulent siphoviruses with genome sizes ~90 kb are grouped within cluster D. The strictly lytic podo- and myoviruses of the genus *Staphylococcus* are distributed within phylogenetic clusters A and C, and feature genomes of either below 20 kb or greater than 120 kb, respectively[17]. However, the true staphylococcal phage diversity has yet to be fully appreciated, as phages were predominantly isolated on *S. aureus* (~90%), and cluster B siphoviruses comprise most of all published phages (>60%). Due to their strictly lytic lifestyles and broader host ranges, cluster A and C phages are commonly exploited for novel therapeutic applications while cluster B phages are attributed a profound role in the evolution of host pathogenicity and gene traffic (reviewed in[18]). In fact, the process of transduction has so far only been deeply demonstrated for some Cluster B phages and one giant staphylococcal myovirus[19]. Those phages facilitate, for instance, the spread of virulence genes on chromosomally encoded gene cassettes[12,20], or promote the dissemination of small plasmids conferring resistance to various classes of antibiotics[21]. However, the phages' exact role in the exchange of genetic material remains unclear as transduction frequencies and their range of influence are still unsettled.

Generally, the host range of staphylococcal phages is thought to be species-specific and restricted at the highest level by the availability of the phage receptor on the bacterial surface. (reviewed in[22]). Wall teichoic acids (WTAs) have here been reported as primary targets[22] and show high intra-species conservation[23] but diversify among different staphylococcal species[24,25]. This evolutionary divergence is postulated to restrict most phage infections and transduction events across the species barrier[26]. Yet, a few reported staphylococcal phages infect diverse species[19,27–29], and a close relationship between CoNS and CoPS phages was described[30–32]. One level below, individual internal defense mechanisms such as restriction-modification systems, CRISPR/Cas, or resident prophages narrow the host range so that a phage can infect some, but never all strains of a species[1,22]. Despite those general tendencies, a well-defined picture of the staphylococcal phage host range is absent, especially when considering non-*S. aureus* species.

In this work, we characterize the host range of 94 phages isolated from a wastewater treatment plant (WWTP). For each member of this phage community, we assess the host range on a diverse set of 117 staphylococci originating from 29 species, and six presumably negative control strains. We show that only four of the 94 isolated phages infect hosts from a single species. Using a network-based approach, we further demonstrate that staphylococcal hosts of different species, ecosystems, or antibiotic resistance phenotypes are closely connected through a multitude of phages. Finally, we reveal that phages bridging these connections have the competence to incorporate foreign genetic material, measured by a chloramphenicol resistance gene. Our findings challenge the commonly reported host specialism of phages and place phages as potent vehicles for bacterial genetic exchange.

## Results

**Host enrichment cocktails unveil a great abundance of staphylococcal phages in wastewater.** To study the host range of staphylococcal phages, we isolated native staphylococcal phages from the influent and effluent of a WWTP in Zürich, Switzerland. Before phage isolation, we assembled five enrichment cocktails of 46 staphylococcal hosts from 17 different species that were selected to produce a diverse community. We assured the growth of each host within an enrichment cocktail by excluding bacteria-harboring cross-infecting prophages or bacteriocin producers (Supplementary Data 1). Using those cocktails, we isolated 155 phages, 134 from the wastewater influent and 21 from the effluent (Supplementary Table 1). Generally, the enrichment hosts spanned types from phage permissive to phage resistant, as we isolated phages on 26 strains from 15 staphylococcal species (Supplementary Data 2). Of the initial 30 CoNS enrichment hosts, 22 strains were phage susceptible, in contrast to only four of the 16 CoPS hosts. This outline resulted in the isolation of 136 phages on CoNS and 19 on CoPS (Supplementary Table 2). Another 24 presumably temperate phages were isolated from wastewater bacterial lysogens by induction. The detection and thus isolation hosts for those phages were mostly *S. epidermidis*, and one *S. sciuri* strain (Supplementary Table 3). Interestingly, those strains also proved highly successful in segregating free phage particles when used in enrichment cocktails (Supplementary Data 2), making them good candidates for further isolation advances.

To sum up, we demonstrate the prevalence of staphylococcal phages in wastewater through the isolation of 179 staphylococcal phages on 15 different staphylococcal species (Supplementary Data 3, Column A−E). As the vast majority of recovered phages (160) were isolated on CoNS (23 strains from 12 species), and the minority (19) on CoPS (4 strains from 3 species) (Supplementary

Table 4, Column A−C), we substantially increase the phage landscape of the genus *Staphylococcus*.

**Most phages have a single isolation host**. To discriminate all isolated phages, and to unravel their host ranges, we selected a diverse host panel of 123 bacteria from 32 different species, including 29 *Staphylococcus* (117), two *Macrococcus* (4), and one *Enterococcus* (2) species. The chosen hosts originated from either the human (40), veterinary (53), or environmental (23) biome and presented a multi-drug resistant (35), resistant (48), or antibiotic susceptible phenotype (40) (Supplementary Data 4). First, the host range served to discriminate between isolated phages, as we considered phages with an equal host range on this array as identical. Phages with identical host ranges on the 123 bacteria were collapsed into cluster groups, and we continued working with only one phage per cluster. This characterization resulted in the discrimination of the 179 isolated phages into 94 unique (Supplementary Data 3, column F). Of these, 80 phages were recovered from the influent and eight from the effluent. Effluent phages that were also isolated from the influent (4) were assigned to the outlet phage fraction. Six different phages further remained from the induction of bacterial lysogens. As we observed an almost 50% redundancy in phage isolation (i.e., from 179 to 94 unique phages), we analyzed the recovery frequency for each phage cluster. To our surprise, 62 out of the 94 unique phages were isolated only once (1 plaque), whereas 32 were isolated between two and 16 times. However, this re-isolation did not arise from an excess of isolation hosts, as 76 phages had only one matching isolation strain, and as few as 10 phages were recovered on more than one species (Supplementary Data 5). Furthermore, the inclusion of multiple enrichment cocktails was beneficial, as > 80% of the enriched phages (77) were isolated from only a single cocktail (Supplementary Data 6). An enrichment process with cocktails of multiple species may raise the question for a selective isolation of broad host rage phages. As only ten of 94 phages were isolated on more than one species, and we applied isolation approaches other than enrichment, we consider not to have particularly selected for broad host range phages. Altogether, our findings highlight the importance of using a comprehensive panel of bacterial hosts for phage enrichment.

**The phage-bacteria incidence matrix is intermediately modular and nested**. Next, we analyzed the individual infection pattern of the 94 distinguished phages (Supplementary Data 4). These data gave rise to a large phage-bacteria interaction matrix with 1,135 positive infection outcomes of possible 11,562 interactions (Fig. 1 and Supplementary Data 7). We measured high-order properties of this phage-host biadjacency matrix, specifically modularity and nestedness. A nested network structure is evoked if phage host ranges build subsets of each other. The most specialized phage infects only the most permissive bacteria, and broader host range phages evolve to infect less permissive hosts without losing the ability to replicate on the ancestor. A recent re-evaluation of phage−host interaction matrices found that phage-bacteria networks are typically nested[7]. Modularity is a characteristic of phage-bacteria infection networks where groups of phages specialize on non-overlapping groups of hosts. It is associated with taxonomy and elicited when a large taxonomic diversity of bacteria is challenged[6]. We expected diverse modules in our interaction matrix, as we impose bacterial hosts of a large species variety and geographic scale.

Supplementary Fig. 1 shows the modularity (a) and nestedness (b) sorting of our sample matrix. We observe an intermediate situation in which neither clear modules nor a strictly nested condition emerged. The calculated nestedness (NODF = 40.91)

is significantly higher than expected from a random matrix ($z$-value $= 47.34$, $\overline{x}_{\text{distribution}} = 21.15$, $95\% = 21.84$, $p = 0.0099$) but sorting has not resulted in a clearly nested structure. Similarly, the calculated modularity $Q = 0.38$ is significantly higher than expected at random ($z$-value $= 47.87$, $\overline{x}_{\text{distribution}} = 0.21$, $95\% = 0.22$, $p = 0.0099$). However, 348/1135 = 30.6% of the interaction occur between the four detected modules. Interestingly, each module consists of host strains from at least three different phylogenetic species groups[13] and four distinct staphylococcal species. Phage permissive hosts from individual species, however, were mostly limited to one module. Only strains from *S. sciuri*, *S. aureus*, *S. haemolyticus*, *S. xylosus*, and *S. lentus* were split between different modules (Supplementary Table 5). We conclude that on the genus level, strains of equal species tend to cluster within a module, whereby individual species do not build modules. Furthermore, the species composition within a module seems unrelated to their phylogenetic relationship. Overall, the observed pattern suggests a limited specialization of staphylococcal phages on individual staphylococcal species.

**Broad host range is a prevailing trait of the isolated staphylococcal phages**. Using the biadjacency matrix, we next sought to analyze phage predation. Overall, isolated phages infected only staphylococcal bacteria. We observed a remarkable high level of infectivity at the species level, as 27 of 29 staphylococcal species were infected. On a strain level, we find an almost equal number of phage-permissive and phage-resistant hosts, with 60 (51.28%) of the 117 challenged staphylococcal strains tolerating phage infection. Consistent with the isolation of phages on mainly CoNS hosts, we observed a clear preference of infection on this bacterial group (89% of all infections). Most challenged CoNS strains showed phage susceptibility (49 of 68 strains), which stands in contrast to the CoPS (11 of 49 strains) (Supplementary Data 8 and Fig. 2). Environmental strains were most permissive (74%), followed by animal (53%) and lastly human (23%) (Table 1 and Supplementary Fig. 2). Nevertheless, the three hosts with the highest phage predation were of animal origin: *S. lugdunensis* I0507 (CoNS), *S. schleiferi* I3823 (CoPS), and *S. epidermidis* I0564 (CoNS) which were permissive for 65, 63, and 58 different phages, respectively. Generally, strains and species were infected by multiple phages, as on average, they were susceptible to $9.2 \pm 15.4$ ($n = 123$) and $22.7 \pm 22.2$ ($n = 32$) different phages, respectively.

Traditionally, staphylococcal phages are reported as species-specific with a narrow host range. Here, we unveil that phages infect $12.0 \pm 5.4$ ($n = 94$) strains from $7.7 \pm 3.7$ ($n = 94$) species on average. In fact, the host range of 90 phages in this natural community spans multiple species, and only four phages exclusively replicated on a single species. Among them, three phages (PG-2021_89, PG-2021_93, and PG-2021_94 on *S. epidermidis*) were isolated by induction and feature a temperate lifestyle (see genomic data below). Hence, their detected plaquing host range may not reflect the true underlying host range. The remaining species-specific phage, PG-2021_6, was isolated from the outlet fraction and plaqued on a single strain (*S. sciuri*). On the other end of the spectrum, PG-2021_17 displayed the broadest lytic potential. This is the sole phage isolated on *S. pseudintermedius*, and infected 32 strains of both CoPS and CoNS from 18 different species. Generally, we find that the host range of most isolated phages (86%) spanned CoPS and CoNS, of which 22 phages covered the two species with the highest pathogenic potential, *S. aureus*, and *S. epidermidis*, and on average, another $10 \pm 2$ ($n = 22$) different species. Furthermore, all 90 broad host range phages infected strains of at least two different staphylococcal species groups[13]. On average, each phage infected strains

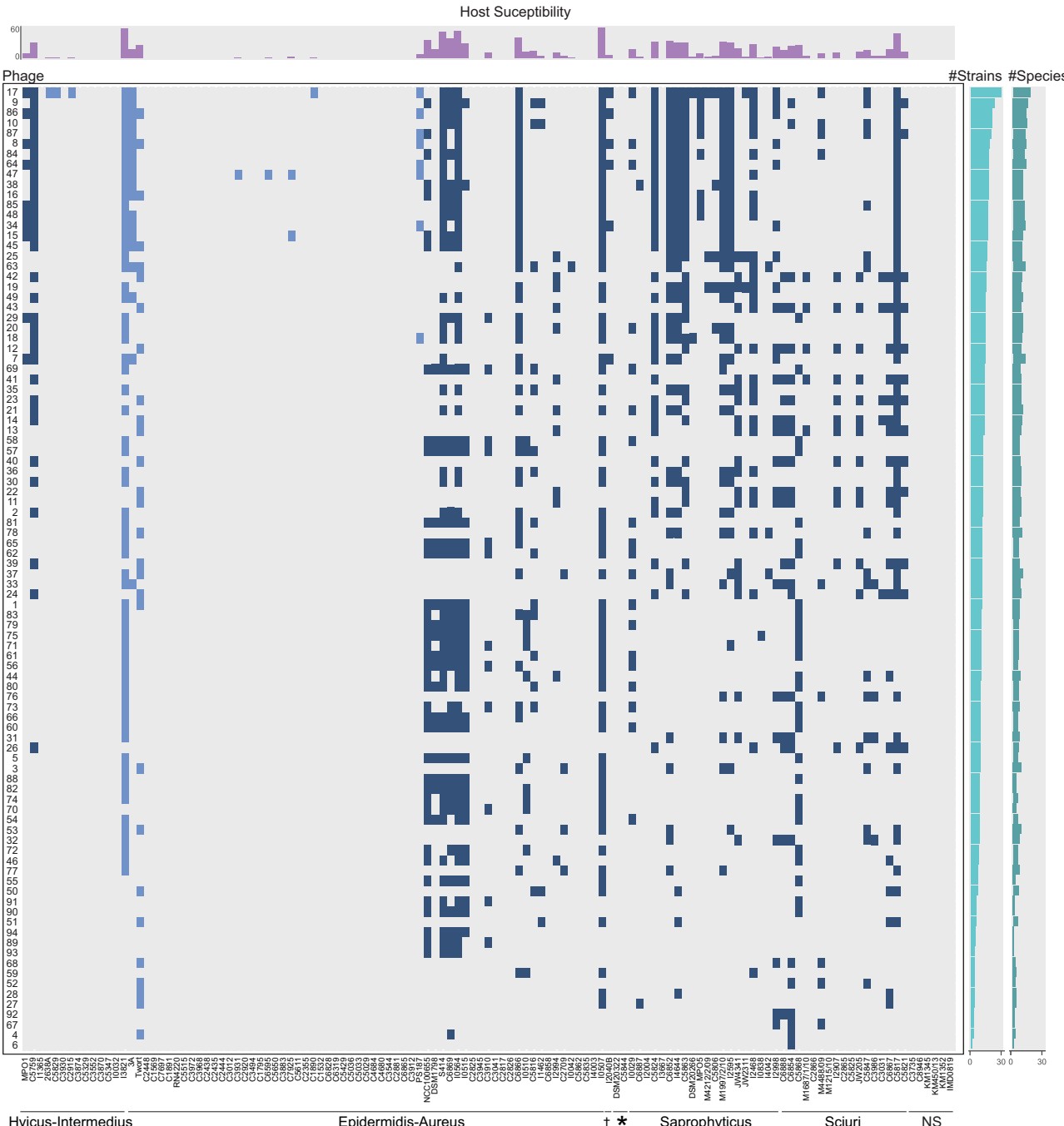

**Fig. 1 Staphylococcal phage-bacteria incidence matrix.** Bacterial lawns of 123 hosts from 32 species were challenged with 94 different staphylococcal phages from wastewater and phage K. Phages on the y-axis are sorted from broad host range to narrow. Bacterial hosts in columns are sorted after cluster-groups and subdivided species as established in[13]. Dagger: Cluster-group Auricularis. Asterisk: cluster-group Simulans. NS: Non-Staphylococcus hosts. Each blue-colored square of the incidence matrix corresponds to a phage-host infection where single plaques were visible. Squares in dark blue indicate infections on CoNS and squares in light blue on CoPS. The phage permissiveness for each host is indicated in the host susceptibility bar chart on top of the incidence matrix, which represents the number of phages infecting a strain. The two bar charts on the right indicate the total number of strains (number of strains, left) and species (number of species, right) a phage infected. The incidence matrix has a diameter of six and a density of 0.1 (=1135/11562). Phages are abbreviated with their final unique numerical identifier (PG-2021_*). Source data are provided as a Source Data file.

from $3.4 \pm 0.9$ ($n = 94$) different species groups and six phages replicated on five out of the six possible groups.

Our findings seem inconsistent with the commonly reported phage specialization[22,33]. However, we hypothesized that specialization does not necessarily contradict a broad multi-species host range, as polyvalent phages can predominantly infect strains of a

single species. A prevalent example is phage K, which is reported as an *S. aureus* phage that replicates on a few other staphylococcal species[27]. On the selected host panel, phage K infected 29 strains from 12 different staphylococcal species. Nevertheless, *S. aureus* hosts (15, 51.7%) predominantly composed its host range (Fig. 3). We evaluated whether a similar proportion of infected strains

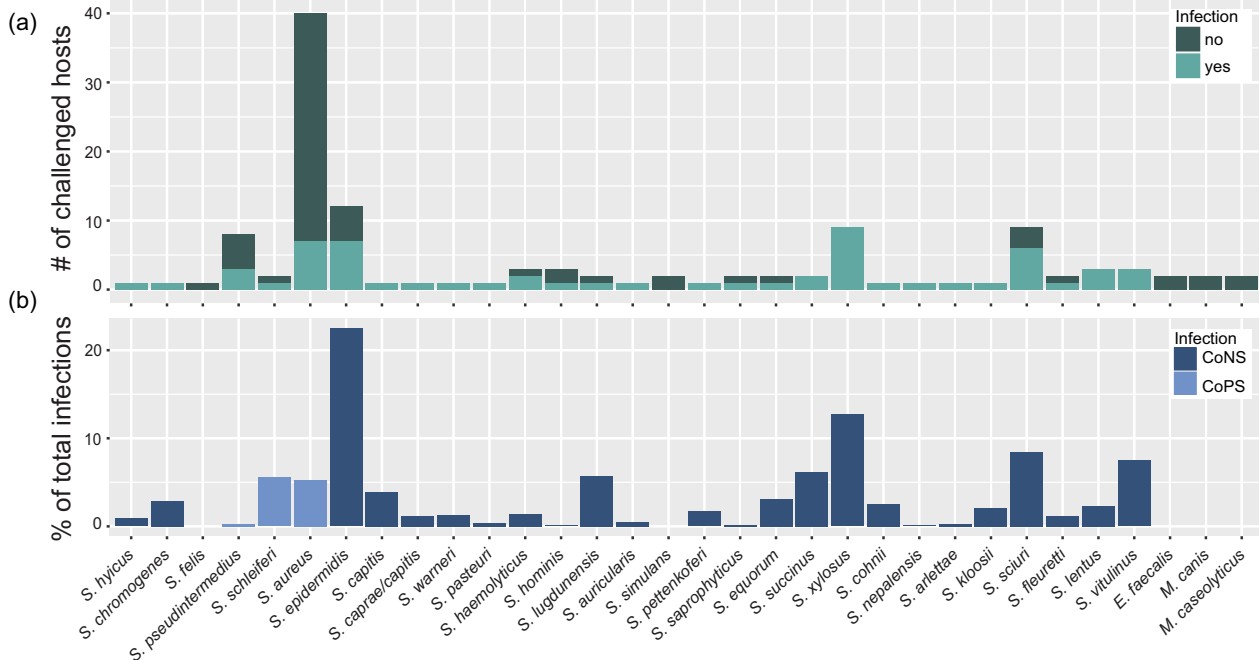

**Fig. 2 Phage infections on staphylococcal species.** Species challenged in the phage-bacteria interaction matrix are shown on the *x*-axis and sorted after the established *Staphylococcus* species groups[13]. **a** For each species, the number of phage resistant and susceptible strains are depicted. **b** Phage infections on each respective species were plotted as a percentage of the total infections detected in the interaction matrix. CoNS coagulase-negative staphylococci, CoPS coagulase-positive staphylococci. Source data are provided as a Source Data file.

**Table 1 General properties of the phage−bacteria interaction network.**

| Network | |
|---|---|
| # Host species (strains) | 32 (123) |
| # Phages | 94 |
| # Interactions (I) | 1135 |
| Size (M) | 11562 |
| Connectance (C = I/M) | 0.098 |
| **Hosts** | |
| Infected species (strains) | 27 (60) |
| Mean (±sd) phage infection per strain | 9.2 ± 15 |
| Mean (±sd) phage infection per species | 22.3 ± 22.5 |
| Maximal phage infections per strain | 65 |
| % infections on CoNS | 89% |
| % infections on CoPS | 11% |
| **Phages** | |
| Maximal species (strains) infection per phage | 18 (32) |
| Mean (±sd) species infections per phage | 7.8 ± 3.7 |
| Mean (±sd) strain infections per phage | 12.10 ± 5.4 |

among the here characterized broad host range phages prevailed. Here, only 30 phages revealed a species tendency, with ≥ 50% of all infected hosts belonging to one individual species. Those phages favorably replicated on *S. epidermidis* (25) and *S. sciuri* (5) (Supplementary Data 9, Column E). On the contrary, the established host range for 60 (64%) of our broad host range phages had no apparent centralization of infection (Fig. 3 and Supplementary Fig. 3).

Our data challenge a strong species tropism of phages within the genus *Staphylococcus* and exclude a harsh species boundary for staphylococcal phages. However, one must consider that the taxonomic diversity of bacteria greatly influences species specificity in each host array. Thus, host range proportions might shift in different collections with an equal number of strains per species.

**Staphylococcal phages infect antibiotic-resistant strains from different biomes.** Phages could be suitable vectors for genetic exchange due to their vast abundance, stability in the environment, and their ability to bridge the spatial separation of donor and recipient bacteria[34]. To assess the possibility of phage-mediated transfer of genetic material across biomes, we first tested the phages' ability to connect hosts from the environmental, veterinary, or human biome. Interestingly, only two phages infected bacteria from a single isolation origin, whereas all other 92 phages infected strains from at least two ecosystems. Within those, the host range of 58 phages connected veterinary and environmental isolated staphylococci and another 34 phages additionally integrated strains recovered from humans (Supplementary Fig. 4a and Supplementary Data 9). Next, we employed the established interaction matrix to examine whether staphylococcal phages replicate on antibiotic-resistant strains, as this conditions their potential to act in the spread of antibiotic resistance. In total, we find 65% of the antibiotic-susceptible, 50% of antibiotic-resistant, and 29% of the multidrug-resistant strains permissive to the tested phages (Supplementary Fig. 2b). Overall, almost half of all infections (44.4%) in the interaction matrix pertained to antibiotic-resistant hosts and similar phage predation occurred between antimicrobial susceptible and resistant hosts (two-sided Wilcoxon rank-sum test with continuity correction, $W = 568.5$, $n = 60$, $p$-value = 0.068; Supplementary Fig. 5). Ultimately, all isolated phages productively infected at least one drug-resistant strain. Our results evidence that diverse staphylococcal phages connect hosts from different ecosystems and drug resistance phenotypes, making them suitable vectors for gene traffic if the phage progenies comprise transducing particles.

**Natural phage communities crosslink species within the genus *Staphylococcus*.** Next, we analyzed the established biadjacency matrix focusing on the interplay between infected bacteria rather than individual phage host ranges. To do so, we reduced the matrix to the 60 phage-permissive strains, which represented 27

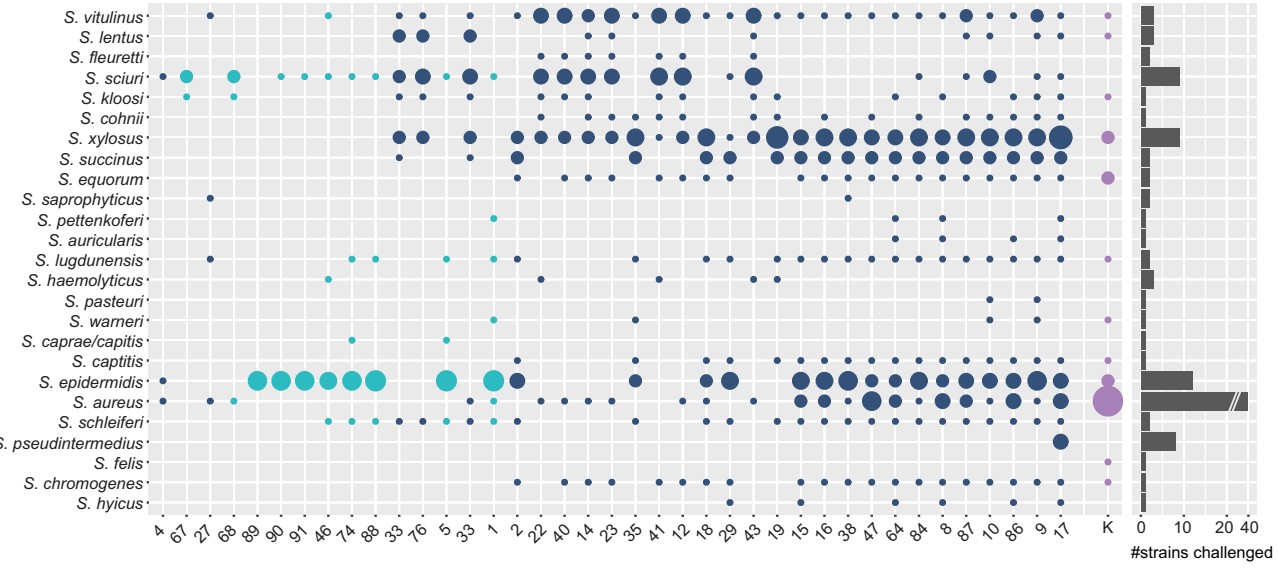

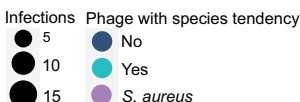

**Fig. 3 Illustration of the host ranges collapsed on the species level for the 40 sequenced phages.** Phages on the *x*-axis are sorted from narrow (left) to broad host range (right). Species on the *y*-axis are sorted after phylogenetic relationship in species groups[13]. A phage host range is depicted as a column, where infection of a staphylococcal species is illustrated using circles. For each respective species, the area of the circle is scaled according to the number of strains a phage can replicate on (scale: 1–15). The total number of strains challenged per species is depicted in the bar-chart on the right. Host ranges on this host array are colored as follows: Phage with species tendency (≥50% of all infections on a single species) in turquoise; phages with no clear species tendency in dark blue; polyvalent phage K in violet. Phages are abbreviated with their final unique numerical identifier (PG-2021_*). Source data are provided as a Source Data file.

different staphylococcal species. The network was collapsed into a bipartite projection in which hosts are represented as nodes and phages as edges. An edge between two bacterial nodes indicates the presence of at least one phage infecting both hosts, and the edges are weighted according to the number of phages that do so. The projection showed an interconnected network with 1,030 host interactions through 93 different staphylococcal phages (connectance = 0.58) (Table 2 and Fig. 4). We sought to establish parameters that best describe how this natural phage community crosslinks members of the genus *Staphylococcus*. On the one hand, we consider the number of shared phages between two hosts as an important marker, as they indicate transfer routes and opportunities for genetic exchange. Thus, the higher the number of shared phages between two hosts, the higher the chance of genetic displacement as multiple phages could govern a transfer. On the other hand, we recognize the number of direct neighbors, which is the count of nodes connected by an edge to the specified node. Neighbors are a measure of centrality and demonstrate the host's impact.

The bipartite projection revealed that staphylococcal strains from different species groups[13] share on average $3.8 \pm 7.3$ ($n = 1293$) phages. Moreover, individual staphylococcal strains and species were connected by $4.3 \pm 7.9$ ($n = 1770$) (Supplementary Fig. 6) and $4.1 \pm 7.5$ ($n = 1666$) different phages, respectively. However, staphylococcal strains from the same species were significantly better connected (two-sided Wilcoxon rank-sum test with continuity correction: $W = 237492$, *p*-value = 1.93 e−15), as on average they were linked by $8.3 \pm 11.4$ ($n = 104$) different phages. Surprisingly, the two best-connected hosts throughout this network belong to different species, species groups[13], and

coagulase types: *S. lugdunensis* I0507 (Epidermidis-Aureus, CoNS) and *S. schleiferi* I3823 (Hyicus-Intermedius, CoPS), which share sensitivity to 58 different staphylococcal phages.

In addition to the numerous transfer opportunities, we found a bacterium to be connected to $34.3 \pm 13.6$ ($n = 60$) strains from $17.4 \pm 5.2$ ($n = 60$) staphylococcal species through phages. The strain with the highest number of neighbors is most likely to receive and donate genetic material. We found *S. vitulinus* C5817 as the most central host that could interact with 56 of 59 available hosts. Furthermore, phage infections connected strains of the species *S. epidermidis* (I0564), *S. lugdunensis* (I0507), and *S. schleiferi* (I3823) to 25 of 26 other staphylococcal species.

Lastly, we appraised the connectivity between ecosystems by phages, as there is a rising fear of genetic mobilization between the human, environmental and animal biome. To do so, we assessed the number of shared phages between hosts of different origins. Surprisingly, we found no significant difference in the average number of shared phages between hosts from the same ($4.7 \pm 8.4$, $n = 565$) or different biome ($4.1 \pm 7.6$, $n = 1205$) (Two-sided Wilcoxon rank-sum test with continuity correction, $W = 1315504$, $p = 0.08981$). Hosts of environmental and veterinary origin, however, were exceptional well connected, as they share $5.7 \pm 8.8$ ($n = 476$) phages on average (Supplementary Fig. 4b). Furthermore, of the 34 neighbors previously found on average for a host in this network, only $11.2 \pm 7.0$ ($n = 60$) share the same isolation biome, whereas $23.1 \pm 9.9$ ($n = 60$) hosts derived from different ecosystems (Supplementary Fig. 7). The interconnection of spatially separated staphylococcal strains becomes critical when addressing the dissemination of drug resistance determinants. Here, we demonstrate that each

**Table 2 General properties of the bipartite network projection. In the bipartite network projection, hosts are nodes and the number of shared phages are weighted edges between nodes.**

| Bipartite network projection | |
|---|---|
| # Host strains (H) | 60 |
| # Host species | 27 |
| # Phages (P) | 93 |
| Size (M = H × (H−1)/2) | 1770 |
| Number of interactions (I) | 1030 |
| Connectance (C = I/M) | 0.58 |
| **Shared phages** | |
| Mean (±sd) between two hosts | 4.3 ± 7.9 |
| Maximum between two host | 58 |
| Mean (±sd) between species | 4.1 ± 7.5 |
| Mean (±sd) within species | 8.3 ± 11.4 |
| Mean (±sd) between environments | 4.1 ± 7.6 |
| Mean (±sd) within environments | 4.7 ± 8.4 |
| Mean (±sd) between drug-susceptible and resistant | 4.3 ± 8.0 |
| **Neighbors** | |
| Mean (±sd) neighbors per strain | 34.3 ± 13.6 |
| Maximum neighbors per strain | 56 |
| Mean (±sd) species neighbors per strain | 17.4 ± 5.2 |
| Maximum species neighbors per strain | 25 |
| Mean (±sd) neighbors per strain from other environments | 23.1 ± 9.9 |
| Mean (±sd) neighbors per strain from the same environment | 11.2 ± 7.0 |
| Mean (±sd) of drug-susceptible neighbors for each drug-resistant host | 16.0 ± 6.7 |

drug-resistant host is connected on average to $16.0 \pm 6.7$ ($n = 33$) drug-susceptible neighbors through $4.3 \pm 8.0$ ($n = 891$) different phages. Our findings unveil phages to connect copious hosts from multiple species, ecosystems, and of different clinical relevance, and place phages as potential vectors for bacterial genetic exchange.

**WWTPs are reservoirs for taxonomically diverse CoNS phages.** We sequenced the genome of 40 viruses of our natural phage community (Table 3) and assessed their morphology by electron microscopy (Fig. 5). Among the 40 sequenced phages, 29 were isolated from the WWTP inlet, seven from the outlet, and four were bacterial lysogens. Overall, we identified 29 myoviral and 11 siphoviral morphologies. Isolated phages from the raw wastewater revealed to be mainly myoviruses (with two siphoviruses), whereas siphoviruses dominated in the treated water. As anticipated, the sequenced myovirus' genome sizes ranged from 128.3 to 145.1 kb, while the siphoviruses separated into two groups between $42.2 - 44.5$ kb and $85.8 - 92.2$ kb[17]. Interestingly, all siphoviruses with a larger genome, thus members of cluster D, were isolated as free viral particles and displayed the distinct morphology with tails > 300 nm. In contrast, the ~ 40 kb siphoviruses were solely isolated after induction (Fig. 5) and lysogeny modules were found only in phages of this cluster. This is coherent with literature, as cluster B siphoviruses are predicted to be temperate, whereas cluster D phages are presumably virulent[17]. Thus, all phages isolated here by induction are siphoviruses of cluster B (Table 3). To date, only three representatives of cluster D siphoviruses are reported. With the characterization of seven novel cluster D phages, we significantly extend the currently available sequencing landscape of this phage fraction. Next, we assigned the closest phage relative for each of our novel phages based on average nucleotide identity (ANI). Interestingly, 29 CoNS viruses shared a relatively high genome identity (>88% ANI) with known staphylococcal phages, while

the other 11 appeared to be distantly related (<70% ANI). We detected a total of 34 tRNAs among 18 phage genomes. All tRNAs-encoding phages corresponded to strictly lytic myoviruses or large siphoviruses. These results are compatible with the hypothesis that tRNAs are more prevalent among virulent phages. They are less well adapted to their replication hosts and hence, have a compositional difference for codon or amino acid usage[35]. Lastly, we computed a phylogenomic analysis using the phage genomes described herein along with 292 staphylococcal phages deposited on NCBI (Fig. 6). As a unique ecosystem, water from a WWTP revealed to contain diverse staphylococcal phages from different families and genera. The phylogenomic tree showed a good agreement between phage morphology, genome length, and taxonomy. However, the extent of the phage host range seemed rather independent, although members of the *Herelleviridae* infected the highest number of strains and species, followed by ~ 90 kb, and lastly, ~40 kb *Siphoviridae* (Fig. 6). In more detail, all visualized myoviruses infected multiple staphylococcal species ($9.8 \pm 3.05$ on average, $n = 44$), while the identified siphoviruses presented a much narrower host range ($2.5 \pm 1.5$ species on average, $n = 12$). Of the four phages that exclusively replicated on a single species, we identified three of them as cluster B phages isolated through induction, and one cluster D phage. Nevertheless, 8 of 12 visualized siphoviruses productively replicated on more than one host species (Supplementary Data 7 and Fig. 5). It is feasible that phages with larger genomes have an extended host range, as they enclose more space to encode arrays of genes that could counteract host defenses. However, one should consider that temperate phages may have a broader host range than observed by productive infection assays. The detection of hosts, in which these phages pursue a lysogenic infection cycle, will expand the here unveiled host-range breath. In conclusion, by sequencing 40 CoNS staphylococcal phages from the same environmental niche, we greatly extend the spectrum of genome diversity. We demonstrate that phages from diverse taxonomic groups infect bacteria from numerous species, ecosystems, and drug-resistant phenotypes within the genus *Staphylococcus*.

**Phages from diverse taxonomic groups encapsidate foreign genetic material.** In this study, we showed the existence of an expansive network among bacteria of different species mediated by staphylococcal phages. Ultimately, we appraised those phages' potential to incorporate foreign genetic material. For this, we transformed a natural *S. sciuri* plasmid pUR2865[36] (3.83 kb) conferring chloramphenicol resistance into *S. epidermidis* S414. This strain was chosen as donor, as it was infected by most sequenced phages (26) and by members of the *Sipho-* and *Herelleviridae*. Prior all experiments, we confirmed the absence of internal prophages in *S. epidermidis* S414/pUR2865 by whole genome sequencing. We propagated phages on *S. epidermidis* S414/pUR2865 and quantified the encapsidated plasmid pUR2865 by qPCR. In addition, generalize transducing staphylococcal phage 80α and myovirus phage K were propagated on *S. aureus* RN4220/pUR2865. The removal of contaminating non-encapsidated DNA was verified using controls as established in[37]. Plasmid numbers ranged from $1.3 \times 10^1$ to $1.6 \times 10^6$ copies/ng phage DNA with high variations between phage samples. Using the detected copy numbers, we estimated the frequency of transducing particle formation. We assumed that transducing particles consist of plasmid multimers only[38], and that as many base pairs of plasmid DNA are incorporated as the respective phage genome length. Figure 7a summarizes the differences in frequencies of phage transducing particles monitored per phage sample. The frequencies of transducing particles harboring the

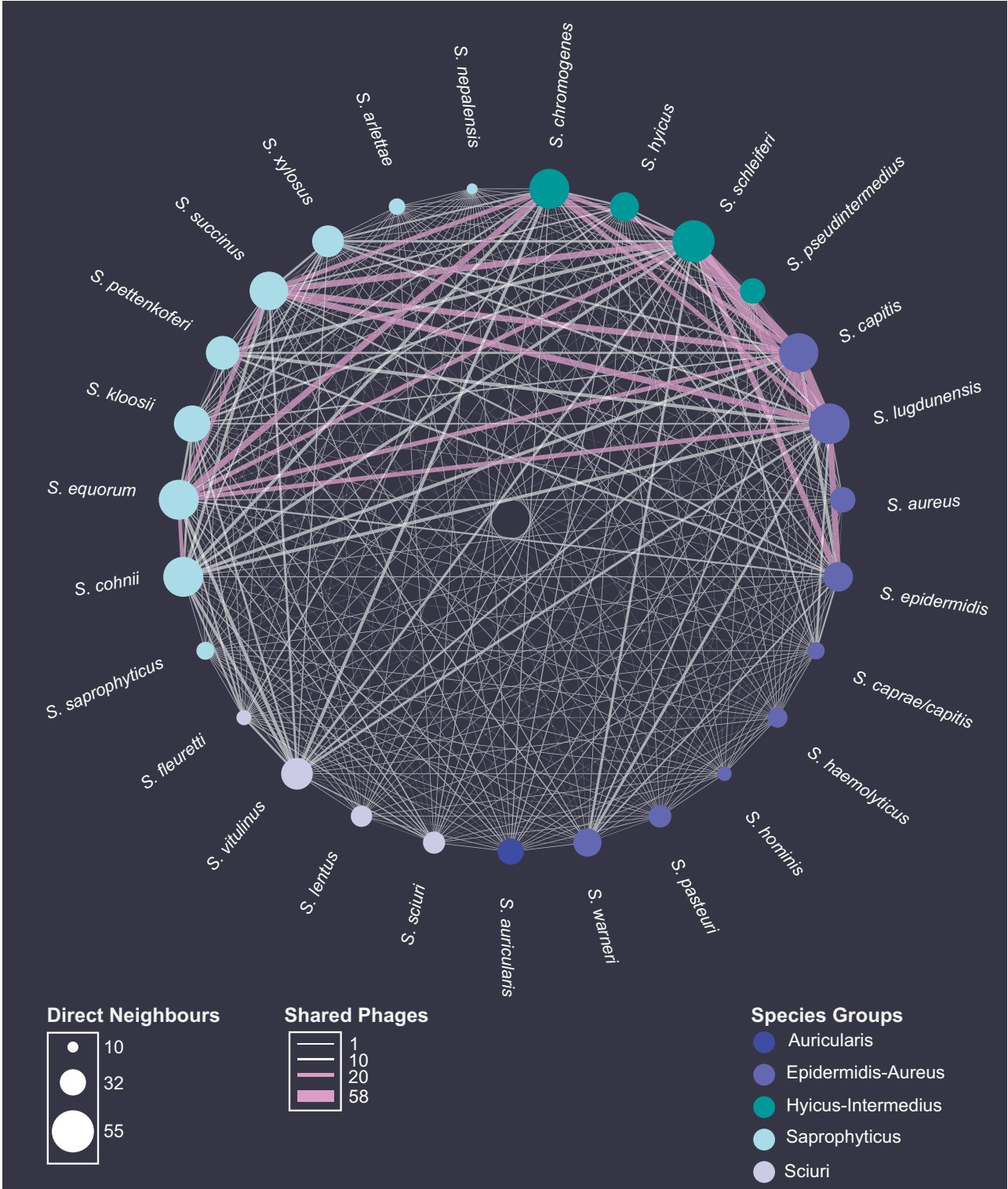

**Fig. 4 Species network with phages as coupling links.** Staphylococcal host species are represented as nodes and sorted after cluster affiliation[13]. The area of each node directly correlates with the average number of strain neighbors a species is connected to. The number of shared phages between species is represented as weighted edges. If >20 phages are shared between two staphylococcal species, edges are colored in pink. Source data are provided as a Source Data file.

plasmid indicate that one out of $1.5 \times 10^2$ to minimal $2 \times 10^7$ phages package foreign genetic material. We expected high plasmid incorporation rates for phage 80α and for cluster B siphoviruses, as transduction ability for those phages is generally accepted[39]. To our knowledge, there is only one report of a

generalize transducing staphylococcal myovirus[19]. Strikingly, with our model, phage 80α showed comparable frequencies ($5 \times 10^{-6} - 7 \times 10^{-8}$) of transducing particles to the here characterized myoviruses. In contrast, the small siphoviruses isolated from bacterial lysogens, and one large siphovirus (PG-2021_46),

**Table 3 Characteristics of the 40 sequenced staphylococcal phage genomes.**

| Phage[1] | ORI[2] | MO[3] | HR[4] | Propagation | Genome (bp) | Termini[5] | %GC | ORFs | tRNA | Closest relative | %ANI |
|---|---|---|---|---|---|---|---|---|---|---|---|
| 1 | I | M | 12/7 | S. epidermidis | 143'764 | DTR (long) | 27.96 | 241 | - | Twillingate | 98.01 |
| 2 | I | M | 13/9 | S. vitulinus | 142'223 | DTR (long) | 30.85 | 244 | 1 | vB_Sau_Clo6 | 92.14 |
| 4 | O | S | 3/3 | S. sciuri | 91'860 | pac | 30.61 | 171 | - | vB_StaM_SA2 | 67.09 |
| 5 | O | S | 10/5 | S. epidermidis | 92'130 | cos | 29.41 | 174 | - | 6ec | 95.45 |
| 8 | I | M | 20/14 | S. equorum | 139'709 | Unknown | 30.94 | 230 | - | vB_Sau_Clo6 | 91.42 |
| 9 | I | M | 25/16 | S. xylosus | 141'528 | DTR (long) | 30.77 | 244 | - | vB_Sau_Clo7 | 92.24 |
| 10 | I | M | 22/15 | S. xylosus | 141'528 | DTR (long) | 30.77 | 270 | - | vB_Sau_Clo8 | 92.12 |
| 12 | I | M | 15/9 | S. vitulinus | 145'091 | DTR (long) | 31.33 | 285 | 3 | vB_SscM-2 | 67.88 |
| 14 | I | M | 14/10 | S. vitulinus | 145'090 | DTR (long) | 31.33 | 245 | 3 | vB_SscM-4 | 69.95 |
| 15 | I | M | 18/11 | S. xylosus | 145'090 | DTR (long) | 31.33 | 285 | - | vB_SscM-5 | 67.83 |
| 16 | I | M | 19/11 | S. xylosus | 141'321 | DTR (long) | 30.80 | 213 | - | vB_Sau_Clo6 | 91.40 |
| 17 | I | M | 32/18 | S. xylosus | 144'971 | DTR (long) | 30.85 | 261 | 2 | vB_Sau_Clo6 | 91.85 |
| 18 | O | M | 15/10 | S. xylosus | 138'844 | DTR (long) | 30.80 | 225 | 2 | vB_Sau_Clo6 | 92.15 |
| 19 | I | M | 16/9 | S. vitulinus | 141'132 | DTR (long) | 31.25 | 247 | 2 | vB_Sau_S24 | 88.86 |
| 22 | I | M | 13/8 | S. sciuri | 144'280 | DTR (long) | 31.33 | 250 | - | vB_SscM-1 | 69.16 |
| 23 | I | M | 14/9 | S. vitulinus | 139'827 | DTR (long) | 28.00 | 236 | - | Twillingate | 97.63 |
| 27 | I | M | 4/4 | S. aureus | 128'279 | DTR (long) | 29.67 | 220 | - | Quidividi | 69.15 |
| 29 | I | M | 15/11 | S. epidermidis | 131'570 | pac | 30.89 | 215 | 1 | VB_SavM_JYL01 | 91.34 |
| 31 | I | M | 10/7 | S. vitulinus | 139'439 | Unknown | 31.59 | 244 | - | vB_SscM-1 | 90.11 |
| 33 | I | M | 12/8 | S. vitulinus | 135'943 | pac | 31.67 | 234 | - | vB_SscM-1 | 90.25 |
| 35 | I | M | 14/9 | S. xylosus | 138'653 | DTR (long) | 30.80 | 254 | - | vB_Sau_Clo6 | 92.19 |
| 38 | I | M | 19/11 | S. epidermidis | 140'647 | DTR (long) | 30.80 | 260 | 1 | vB_Sau_S24 | 92.47 |
| 40 | I | M | 13/8 | S. vitulinus | 142'875 | DTR (long) | 31.35 | 270 | 3 | vB_SscM-2 | 69.35 |
| 41 | I | M | 13/7 | S. sciuri | 145'090 | DTR (long) | 31.34 | 244 | 3 | vB_SscM-1 | 68.67 |
| 43 | I | M | 15/9 | S. sciuri | 145'090 | DTR (long) | 31.34 | 287 | 3 | vB_SscM-1 | 67.88 |
| 46 | O | S | 8/5 | S. epidermidis | 86'018 | DTR (short) | 29.66 | 152 | - | 6ec | 94.90 |
| 47 | O | M | 19/11 | S. xylosus | 142'885 | DTR (long) | 30.70 | 238 | - | vB_Sau_Clo6 | 92.32 |
| 64 | I | M | 19/14 | S. succinus | 142'287 | DTR (long) | 30.89 | 231 | 1 | vB_Sau_Clo6 | 91.52 |
| 67 | I | S | 3/2 | S. sciuri | 92'064 | cos | 30.57 | 189 | 2 | vB_StaM_SA2 | 66.72 |
| 68 | I | S | 4/3 | S. sciuri | 91'947 | cos | 30.61 | 197 | 1 | vB_StaM_SA2 | 66.92 |
| 74 | O | S | 9/5 | S. epidermidis | 85'762 | pac | 29.66 | 150 | 1 | 6ec | 95.62 |
| 76 | I | M | 10/6 | S. vitulinus | 139'439 | Unknown | 31.59 | 245 | - | vB_SscM-1 | 90.13 |
| 84 | I | M | 19/12 | S. xylosus | 139'439 | DTR (long) | 31.59 | 246 | 2 | vB_Sau_S24 | 92.22 |
| 86 | I | M | 22/14 | S. xylosus | 141'291 | DTR (long) | 30.76 | 228 | 2 | vB_Sau_Clo6 | 90.43 |
| 87 | I | M | 21/13 | S. xylosus | 141'212 | DTR (long) | 30.75 | 239 | - | vB_Sau_Clo6 | 92.65 |
| 88 | O | S | 9/4 | S. epidermidis | 92'222 | DTR (long) | 30.83 | 174 | 1 | 6ec | 95.14 |
| 89 | ID | S | 5/1 | S. epidermidis | 43'039 | Unknown | 35.11 | 77 | - | IME1348_01 | 95.03 |
| 90 | ID | S | 6/2 | S. epidermidis | 44'493 | Unknown | 34.72 | 65 | - | IME1348_01 | 94.72 |
| 91 | ID | S | 6/2 | S. epidermidis | 42'188 | Unknown | 34.97 | 61 | - | IME1348_01 | 95.82 |
| 93 | ID | S | 4/1 | S. epidermidis | 43'459 | Unknown | 34.37 | 79 | - | SepiS-phiIPLA7 | 96.16 |

[1]Phages are abbreviated with their final, unique numerical identifier (PG-2021_*).
[2]Isolation origin (I: inlet, O: Outlet, ID: Induced).
[3]Morphology (M: Myovirus, S: Siphovirus).
[4]Host range as number of strains/species infected.
[5]Termini (DTR: direct terminal repeats).

showed particularly high frequencies between $6.6 \times 10^{-3}$ and $1.6 \times 10^{-5}$. These suggest a more targeted packaging approach of foreign genetic material. Thus, we predicted the phage genome termini in silico, which should reflect the phages' DNA packaging mechanism (Table 3 and Fig. 7b). Interestingly, in several cases, predicted packaging mechanisms did not correlate with phage morphology, and we find high encapsidation frequencies for phages with other packaging mechanisms than the previously found transducing pac[40,41] or cos[42,43] phages. Yet, a pac mechanism is likely for the four induced small siphoviruses with high encapsidation rates, as PhageTerm predicted terminally redundant and circularly permuted genome ends. However, due to a low statistical signal, a definitive confirmation was not obtained.

Our results confirm that plasmid-borne genetic material can be used by phages for mobilization. Furthermore, we demonstrate that multiple phages from diverse taxonomic groups package foreign genetic material, albeit at various frequencies. These data impose great potential for phage-mediated genetic transfer among bacteria, supported by the fact that phages are involved

in far more numerous microbial connections than previously assumed.

## Discussion

Earlier studies have addressed the staphylococcal phage host range to predict the therapeutic fitness of phages or their impact on staphylococcal host diversity. However, phage-host arrays were mostly limited to a small number of species[44]. In fact, most studies focused on phages from S. aureus[27,45–48], and only a few included phages infecting CoNS[49–52]. This restricted variety impeded broad conclusions and lead to an underestimated breath of host range for staphylococcal phages. Our data contains an unprecedented diversity, as it comprises almost 12,000 separate attempts to infect 123 hosts from 32 species with 94 different staphylococcal phage isolates. Using this phage-bacteria interaction matrix, we provide evidence that a broad host range is a dominant trait among the here isolated staphylococcal phages. Interestingly, the ability to infect strains across the species barrier

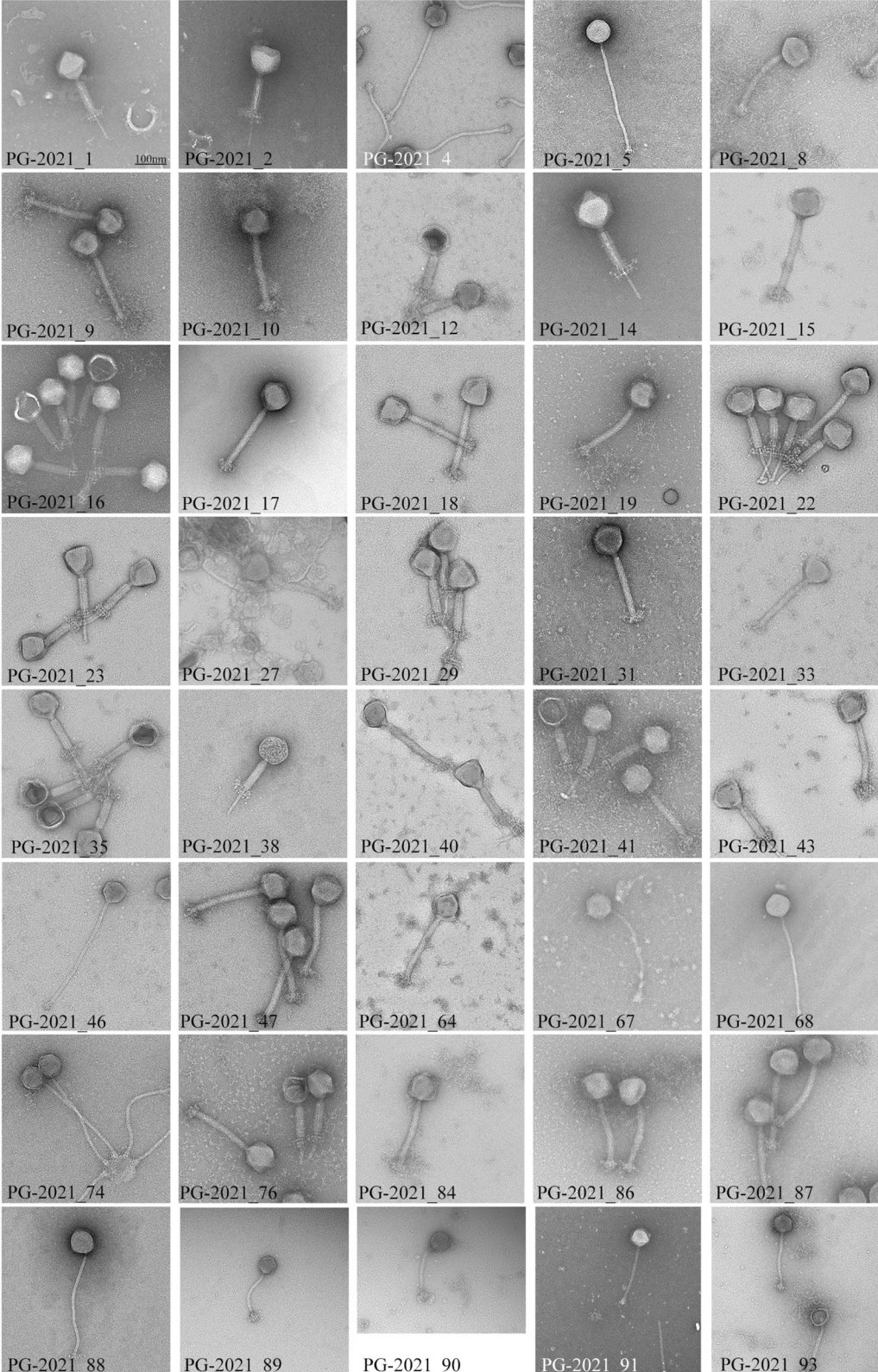

**Fig. 5 Electron micrographs of the sequenced staphylococcal phages.** Phages were isolated from the wastewater treatment plant inlet, outlet, or by induction of bacterial lysogens. For each phage, at least 10 representative pictures were taken. All pictures are adjusted according to the displayed scale-bar in the top-left picture.

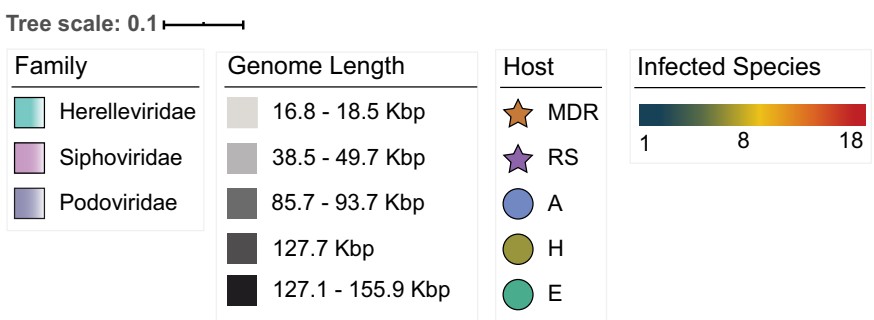

**Fig. 6 Phylogenomic tree of staphylococcal phages.** All published staphylococcal phages are displayed together with the here isolated and sequenced CoNS-infecting viruses. For each phage genus in brackets, a representative phage is indicated. The host range of the here characterized phages is represented as follows: the number of infected species is indicated using a continued color scale; isolation origin, and antimicrobial-resistant phenotype of infected hosts are represented using colored circles and stars, respectively. Phages infect A: hosts isolated from animals, H: hosts isolated from humans, E: hosts isolated from the environment. MDR: host is multidrug resistant. RS: phage infects hosts with antimicrobial-resistant and susceptible phenotypes.

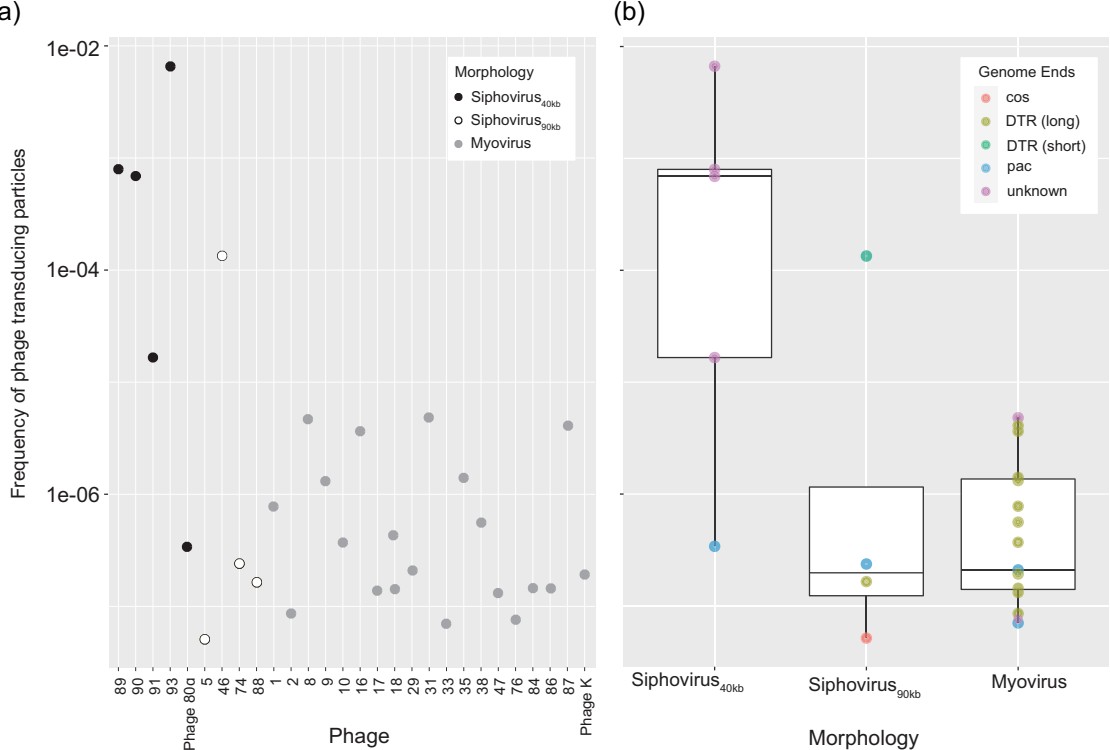

**Fig. 7 Frequency of transducing particles for diverse staphylococcal phages. a** Estimated frequency of transducing particles for each respective phage and corresponding phage morphology. Phages are abbreviated with their final unique numerical identifier (PG-2021_*). **b** Mean (+ sd) frequencies of transducing particles for each phage morphology, i.e., 40 Kb Siphovirus ($n = 5$ phages), 90 Kb Siphovirus ($n = 4$), Myovirus ($n = 19$). Phage termini were predicted using PhageTerm[100] and are illustrated using colors. DTR: direct terminal repeats. Data are represented as boxplots where the middle line is the median, the lower and upper hinges correspond to the first and third quartiles (the 25th and 75th percentiles), the upper whisker extends from the hinge to the largest value at most than 1.5 X IQR from the hinge (where IQR refers to the range between the first and third quartile) and the lowest whisker extends from the hinge to the smallest value no further than 1.5 X IQR, while data beyond the end of the whiskers are outlying points (**b**). All datapoints are plotted individually. Source data are provided as a Source Data file.

and hosts from different ecological and clinical backgrounds was not restricted to a specific phage group. On the contrary, phages with both myo- and siphoviral morphology, as well as temperate and virulent lifestyle, presented this trait. Therefore, we argue our findings to challenge the notion of a strong species tropism within the genus *Staphylococcus*[22,33], and confront the assumption that differences in WTA structure restrict phage infection across species. We hypothesize that WTA structures might not be that evolutionary divergent[53]; phages might recognize conserved structures on different WTAs, or phages might recognize alternative, more conserved receptors on the bacterial cell wall[52]. Nevertheless, we need to question whether a broad host range pattern prevails in general for phages within the genus *Staphylococcus*, as phages from other sources or geographic locations could behave differently. While the infection of a broad spectrum of hosts is desirable for phages in therapy, it simultaneously implies opportunities to spread genetic material. Indeed, phage-mediated HGT is considered one of the primary driving forces for the spread of antimicrobial resistance in staphylococci[15]. However, it is thought to occur rarely, and primarily within species due to estimated narrow host ranges[54]. Using a bipartite network, we demonstrate that multiple phages are shared between antimicrobial-resistant and susceptible hosts, and that each drug-resistant host in this network is, on average, connected to 16 drug susceptible neighbors. We quantified bacterial DNA encapsidation rates for 19 myoviruses and 9 siphoviruses from this network and detected packaged plasmid DNA in all assessed phages, confirming this competence as widespread among the characterized staphylococcal phages[19,39,41]. Our data indicate that one

phage particle out of every hundred to minimal $10^7$ phage particle is transducing. However, those numbers do not necessarily reflect the frequency of generalized transduction due to the following reasoning. We propose that within phage transduction one must acknowledge two main bottlenecks. First, the capability of phages to incorporate foreign DNA and at which frequency transducing particles are being formed. This is dependent on individual phage characteristics, and on type and location of the bacterial cargo DNA within the host. The second bottleneck applies to the delivery and expression of the cargo DNA in the cytoplasm of the recipient bacteria. This can highly differ between strains, as it is mostly depending on the bacterial "immune system", such as restriction-modification systems and CRISPR-Cas[1]. In simplified models, studies propose that transduction efficiencies, thus the successful delivery and expression of cargo DNA in a recipient bacterium, is approximately 3%[40,41]. To this regard, upcoming studies will determine the ability of the here detected transducing particles to spread the drug resistance element across this unique network.

In conclusion, this study reveals an expansive interspecies communication network and place phages as central mediators for bacterial connectivity. Our findings support the speculated interspecies horizontal transfer of adaptive genetic material by phages and exemplify the impact of phage populations on the evolution of human pathogens.

Addendum in Revision: During the time this paper was prepared for publication, the taxonomy of the *Staphylococcaceae* family has been revised. Following phylogenomic analyses, Madhaiyan et al.[55] propose species belonging to the *S. sciuri*

species group (relevant here *S. sciuri*, *S. fleurettii*, *S. lentus*, and *S. vitulinus*) be reassigned to the novel genus *Mammaliicoccus*. Furthermore, *S. schleiferi* subsp. *coagulans* was assigned a novel species *S. coagulans*.

## Methods

**Wastewater sampling.** Water samples were collected at the WWTP in Au (Zurich, Switzerland) on July 24th, 2018. The WWTP Rietliau receives 7–30 million L of wastewater a day from industry and private households and processes it within 24 h. Water is treated with primary and secondary treatments; no tertiary disinfection process is applied. However, half of the treated water is filtered through a 0.035 μm membrane (ZeeWeed hollow fiber membranes technology) prior the release into Lake Zurich. Samples (2.5 L each) were taken at the entrance after the mechanical clearance and from the effluent. Both inlet and outlet samples were centrifuged at 17,000 × *g* for 30 min at 4 °C. The supernatants were 0.22 μm PES sterile filtered and kept at 4 °C for phage enrichment. The bacterial pellets were suspended in 20 ml 0.85% NaCl and held at −20 °C for prophage induction.

**Bacterial strains, culture conditions, and plaque assay.** Bacterial strains for this study[46,56–89] were seeded on tryptic soy agar (TSA, 2% agar and 30 g L⁻¹ tryptic soy broth (TSB)) and grown in TSB overnight at 37 °C. Plaque assays were carried out using LC agar as top agar (10 g L⁻¹ casein peptone, 5 g L⁻¹ yeast extract, 128 mM NaCl, 55.5 mM glucose, 2 mM MgSO₄, 10 mM CaCl₂, 0.4% agar), and TSA as bottom agar. Phages (10 μL, serially diluted) were mixed with bacterial hosts in molten soft agar (47 °C), plated, and incubated overnight before quantification. For spot assays, bacteria were inoculated into molten soft agar (47 °C), plated, and phage concentrates (5 μL, serially diluted) were dropped onto.

**Enrichment cocktail constitution.** Five cocktails were generated to enrich staphylococcal phages from wastewater. Staphylococcal strains for each cocktail were selected to produce a diverse community and combined either randomly (cocktail A), or according to their origin (cocktail B: animal-related strains; cocktail C and D: environmental isolated strains; cocktail E: lab strains). To assure growth harmony for each bacterium within a cocktail, strains with cross-infective prophages or bacteriocin producers were excluded. For this, all 76 selected strains were induced using Mitomycin C and UV irradiation (protocol adapted from[90]). Briefly, 50 μL of a fresh overnight culture was inoculated in 5 mL TSB and incubated on a shaker for 2 h at 37 °C. The initial absorbance was measured at OD₆₀₀. Mitomycin C was added to a final concentration of 0.5 μg mL⁻¹, and bacterial suspensions were shaken at 37 °C. For UV irradiation, cells were centrifuged at 6,000 × *g* for 10 min at room temperature. The pellet was resuspended in 5 mL 0.1 M MgSO₄ and irradiated with UV-Light (2400 μJ cm⁻²). After irradiation, cells were transferred to double strength TSB, protected from light, and incubated on a shaker at 37 °C. The absorbance of both UV and Mitomycin C induced strains was then measured every hour for 6 h or until a decrease of the OD₆₀₀ was observed. The bacterial cultures were then centrifuged at 3,000 × *g* for 12 min at 4 °C, the supernatant 0.22 μm sterile filtered, and stored at 4 °C. For all induction experiments, *S. aureus* Newman served as a positive control, as it contains three inducible prophages that lyse *S. aureus* RN4220[91]. Spot assays were performed to assess the presence of cross-reactive phages that interfere with the growth of strains within a cocktail.

The radial streak method was applied to determine whether cocktail members restrain the growth of others by the production of bacteriocins or other extracellular antimicrobial compounds (protocol adapted from[92]). In short, the area of a small circle was inoculated with a 0.5 McFarland bacterial suspension of each cocktail candidate member in the center of a fresh plate. The plates were incubated at 37 °C for 24 h, and all remaining members of the respective cocktail (0.5 McFarland) were then radially streaked from the border of the dish to the circle area. If the central bacterial strain provoked a zone of growth inhibition after a second incubation, it was excluded.

In summary, out of the 76 staphylococcal strains tested, 25 interfered with bacterial multiplication of selected cocktail members through either inducible prophages (19) or bacteriocin production (6). Five additional strains were removed because of a low growth rate. Based on the remaining 46 strains, we established five enrichment cocktails (A-D), each consisting of eight to eleven staphylococcal hosts from different species (17), origins (19 environmental, 14 animal, three humans, and ten unknown), and coagulase groups (16 CoPS and 30 CoNS) (Supplementary Data 1).

**Phage enrichment and isolation.** Inlet and outlet phage suspensions were enriched for staphylococcal phages using the five constituted enrichment cocktails independently. For each cocktail and sample, 80 mL of the viral suspension was supplemented with 20 mL 5 × TSB and 100 μL of a fresh overnight culture of every cocktail member. The ten suspensions were then incubated overnight at 37 °C. After this first round of enrichment, viral suspensions were centrifuged at 17,000 × *g* for 30 min at 4 °C, and the supernatants 0.45 μm PES sterile filtered. For the second enrichment, 20 ml of 5 × TSB and 100 μl of a fresh overnight culture of the same cocktail members were added anew and processed as described above. The

enrichment process was repeated for a total of three rounds. Staphylococcal phages were detected by spotting 10 μl of the enriched viral suspensions on a bacterial lawn of each enrichment host, and plates were incubated overnight at 37 °C. If a zone of lysis or individual plaques were visible the next day, a plaque assay was performed with serially diluted phage suspensions. Plates with single lysis plaques were examined for different plaque morphologies, and a maximum of three were picked for phage purification for each plate. Phages were purified by repeatedly plating and picking individual plaques for three rounds.

For prophage induction and isolation, bacterial pellets frozen from wastewater were thawed and resuspended in 20 ml double strength TSB supplemented with 6.5% NaCl for staphylococcal enrichment. After overnight incubation, 10 ml of each enrichment was added to 490 ml TSB, and the initial absorbance (OD₆₀₀) was measured. Cells were grown until an OD₆₀₀ of 0.5, and the sample split for the induction with Mitomycin C or UV irradiation. For Mitomycin C induction, a final concentration of 1 μg mL⁻¹ was added, and the suspension was incubated at 37 °C for 6 h. For UV irradiation, cells were centrifuged at 6,000 × *g* for 10 min and the pellet resuspended in 125 mL 0.1 M MgSO₄. This resuspension was irradiated (4400 μJ cm⁻²), transferred to 125 ml double strength TSB, protected from light, and was incubated for 6 h at 37 °C. Finally, induced samples were centrifuged at 10,000 × *g* for 15 min at 4 °C, the supernatants 0.22 μm PES sterile filtrated, and stored at 4 °C. For temperate phage detection, serially diluted phage suspensions were dropped on all hosts selected for host range determination (Hosts in Supplementary Data 6). If either a zone of lysis or individual plaques were visible after overnight incubation, phages were picked and purified as described above.

**Phage host range determination.** Phage host ranges were assessed on 123 strains (32 species) that originated from human (40), veterinary (53), or environmental settings (23) harboring a multidrug-resistant (35), resistant (49), or antibiotic susceptible phenotype (40). The hosts were chosen to represent a diverse community of both CoNS (68) and CoPS (49), as well as other Gram-positive bacteria (6) (Supplementary Data 4). For the classification of multidrug-resistant strains, bacteria resistant to three or more antibiotic families were considered multi-drug resistant, whereas the Macrolide-lincosamide-streptogramin B (MLSᵦ) resistance phenotype was classified as one family. Staphylococcal strains with an unknown coagulase phenotype were assessed for coagulase production using Staph Rapid Latex Test Kit (Brunelli, #271060). Each phage lysate was spotted (5 μl) in duplicates at five concentrations (10⁸–10⁴ pfu mL⁻¹) onto those selected hosts. If single lysis plaques appeared in any dilution after overnight incubation, the strain was considered susceptible to the respective phage, and an infection event was reported. Lysis from without (LFW) events, where a bacterial lysis halo without single visible plaques appears, were additionally reported but not considered as infection. *Staphylococcus* phage K propagated on *S. aureus* PSK ATCC 19685 was used as a reference for all host range assays. Phages with equal host ranges on all 123 hosts were clustered, and further characterizations were continued with one selected phage per cluster.

**Phage propagation.** Phages were produced using the double-agar-layer method and washed off 20−80 semi-confluent lysis plates using SM buffer (200 mM sodium chloride, 10 mM MgSO₄, 50 mM tris, and 0.01% gelatin, pH 7.4) and gentle agitation for 4 h. The phage lysates were collected, and cellular debris or agar remnants were removed by centrifugation at 5,000 × *g* for 10 min at 4 °C. The supernatant was 0.22 μm sterile filtrated. Phage particles were precipitated with 7% PEG₈₀₀₀ supplemented with 1 M NaCl in ice water for two days. The precipitated phages were collected by centrifugation at 10,000 × *g* for 20 min at 4 °C, and pellets were dissolved in 8 mL SM buffer. Phages were purified by CsCl ultracentrifugation. Briefly, the density of each phage suspension was adjusted 1.15 g mL⁻¹ using CsCl and added on top of a three-layer (1.7, 1.5, and 1.35) CsCl density gradient. The gradient was centrifuged at 82,000 × *g* for 2 h at 10 °C, and the phages were collected between the 1.35 and 1.5 density layers. All purified phages were dialyzed overnight at 4 °C in 4 L SM buffer (50 kDa cut off) under gentle magnetic stirring.

**Phage DNA extraction.** Phage DNA was extracted using the phenol/chloroform DNA extraction method. In short, 640 μL of propagated phage lysate (> 10¹⁰ pfu mL⁻¹) were treated with 10 U DNase I for 1 h at 37 °C, and the enzyme heat-inactivated for 10 min at 65 °C in the presence of 20 mM EDTA. Proteinase K was added to a final concentration of 100 μg mL⁻¹, the sample vortexed and incubated for 1 h at 50 °C with gentle agitation. Next, one volume of phenol:chloroform:isoamyl alcohol (25:24:1) was added, the sample centrifuged for 13'000 × *g* for 15 min, and the aqueous layer extracted. This step was repeated with 1 volume chloroform:isoamyl alcohol (24:1). DNA was precipitated by adding 50 μl 5 M NaCl and 0.7 volumes of isopropanol. The next day, the DNA was pelleted with 13'000 × *g* for 20 min at 4 °C, and the pellet washed twice with ice-cold 70% EtOH. DNA was resuspended in 50 μl 10 mM Tris (pH = 8.0), and the concentration was measured using Qubit.

**Electron Microscopy.** Propagated phages (≥ 10⁹ pfu mL⁻¹, 8 μl) were let absorb to negatively discharged (45 s, 3 × 10⁻¹ mbar, 25 mA) carbon-coated copper grids (Quantifoil) for one minute. Grids were washed twice in pure water and adsorbed particles negatively stained for 20 s with 2% uranyl acetate or phosphotungstic acid.

They were observed at 100 kV on a Hitachi HT 7700 scope equipped with an AMT XR81B Peltier cooled CCD camera (8 M pixel). In total, 56 phages were assessed for their morphology using electron microscopy (Supplementary Data 7). Phage morphologies for all sequenced isolates are displayed in Fig. 5.

**Genome library preparation, sequencing, and bioinformatics**. Forty phages were selected for whole-genome sequencing. Precedence was given to phages obtained from the WWTP outlet and bacterial lysogens, and later those that infected hosts from diverse ecosystems and with different drug-resistant phenotypes. Phage genomes were Illumina sequenced if genomic DNA yields were <1 µg. For Illumina sequencing, multiplexed libraries were prepared using the Illumina Trueseq Nano library prep according to manufactures' instructions. Phage DNA was paired-end sequenced with 0.5 million reads (150 bp read$^{-1}$) using the MiSeq sequencer. Raw reads were trimmed with Trimmomatic[93] in default settings and assembled using SPAdes[94] in careful mode. For Pacbio sequencing, gDNA (ca. 1 µg) was mechanically sheared to the average size distribution of 8–10 kb, using a Covaris gTube (Covaris p/n 520079). Multiplex libraries were prepared using the SMRTBell™ Barcoded Adapter Complete Prep Kit–96, following the manufacturer's instructions (100–514–900, Pacific Biosciences). Tagged libraries were sequenced in a 1 M SMART Cell with PacBio Sequel. De-multiplexed reads were assembled using the Hierarchical Genome Assembly Process[95] (HGAP4, SMRT Link and Tools v8.0.0). When needed, Sanger sequencing was used to close gaps in the assembled genomes. Open reading frames (ORFs) were predicted with prodigal[96], and annotated using multiPhATE[97] with blastn against the NCBI virus, blastp against pVOGs[98], PhAnToMe, and NCBI virus, and jackhmmer against the pVOGS database. Potential tRNAs in phage genomes were predicted using tRNAScan-SE v2.0.5[99]. Phage termini were predicted using PhageTerm[100].

**Phylogenetic analysis**. Bio.Entrez package from biopython within a conda environment was used to retrieve fully sequenced staphylococcal phage genomes deposited at GenBank as of June 2020 ($n = 292$)[17]. Unverified cRNA or partial phage genomes were excluded from the analysis. The closest relative on NCBI was determined by average nucleotide identity (ANI) values. For this, genome-to-genome comparison of all staphylococcal phage genomes was performed using JSpecies[101,102]. For phylogenomic analysis, an all-versus-all comparison of the coding DNA sequences derived from the viral genomes was performed through Diamond[103] (more sensitive mode, identity ≥ 30%, bitscore ≥ 30, alignment length ≥ 30 amino acids, and $e$ value ≤ 0.01). Distances between genomic sequences were calculated using the Dice metric where: $DAB = 1 - (2 × (AB)/(AA + BB))$, where AB is the bitscore sum of all the valid protein matches of sequence A against sequence B, while AA and BB are the bitscore sum of all the valid protein matches of sequence A against itself and of all the valid protein matches of sequence B against itself, respectively. The obtained Dice distances matrix was used to build a phylogenomic tree through neighbor-joining algorithm[104] implemented in the Phangorn package of R. The obtained tree was visualized through iTOL[105].

**Analysis of nestedness and modularity**. Modularity and nestedness were calculated based on the generated biadjacency matrix. All bacteria resistant to phage infections ($n = 63/123$) were removed from the dataset, resulting in a 60 bacteria × 94 phage matrix. Modularity (Q) was calculated with the Ipbrim R package[106] using the findModules function with 100 iterations. Nestedness was measured with the nestednodf function of the vegan package[107] in R (Supplementary Data 10). The null mode method "r00" was chosen as it preserves the matrix size and number of interactions. Statistical significance was evaluated using the oecosimu function with 100 simulations (one-sided testing with statistic assumed greater than simulated values).

**Network analysis**. The network analysis was based on the host range matrix consisting of 123 bacterial hosts and 94 phages isolated from wastewater. A binary incidence matrix was generated from the data in which infections are indicated as one, and no interaction is marked as zero. Phage-resistant hosts ($n = 63/123$) were removed, and a bipartite network was generated using the R package igraph[108] (Supplementary Data 10). In this network, phage permissive bacteria (60) and respective phages (94) were represented as nodes where an edge between a bacterial and phage node indicates phage infection. This network was further collapsed into a bipartite projection, in which only bacteria are represented as nodes and phages as edges connecting two bacterial nodes. The number of shared phages between two nodes was assigned as an edge attribute. Best connected hosts were identified by the highest number of shared phages. The mean number of shared phages was calculated by averaging the values for all possible host pairs ($M = 60 × 59/2 = 1770$) in the bipartite projection. Host pairs with no shared phages (740) were included in the average with a value of zero. Direct neighbors (degree of a node) were counted as the sum of all nodes that are connected by an edge to a specified node. For a subset neighbor count, only neighbors with a specific attribute like resistance profile, origin, or species affiliation were considered.

**Encapsidation rates**. A subset of sequenced staphylococcal phages was assessed for their ability to encapsidate foreign genetic material. For this, the donor S. epidermidis S414, susceptible to most sequenced staphylococcal phages (26 out of 40) was selected. A small, natural occurring S. sciuri plasmid pUR2865 (3.83 kb)[36] conferring resistance to chloramphenicol ($cat_{pC221}$) was chosen as genetic marker for encapsidation. The plasmid was transformed into S. epidermidis S414 and S. aureus RN4220 using 1 µg of plasmid DNA and a pulse at 2.0 kV 100 Ω, 25 µF, or 1.8 kV,600 Ω, 10 µF, respectively[109]. To rule out the presence of internal prophages in S. epidermidis S414/pUR2865, this strain was sequenced with Illumina (NEB-Next Ultra II DNA Library preparation kit according to manufactures instructions, MiniSeq sequenced, 150 bp paired end read configuration), assembled with SPAdes[94], and searched for viral sequences using VIBRANT[110]. No viral sequences were detected in the assembly. Sequenced phages infecting S. epidermidis S414/pUR2865 were then propagated on this strain. Equally, staphylococcal phage K and phage 80α were propagated on S. aureus RN4220/pUR2865. Phage particles were washed off three semi-confluent lysis plates using SM buffer and supernatants were 0.22 µm sterile filtrated. Phage lysates were purified using CsCl density gradient centrifugation and dialyzed (see Phage Propagation for details). Samples (620 µl) were treated with 100 Units DNase I and phage encapsidated DNA was extracted (see Phage DNA Extraction for details). DNA concentrations were measured in duplicates using Qubit Fluorometric Quantification (Thermo Fisher Scientific). Copy numbers of the chloramphenicol resistance marker $cat_{pC221}$ on pUR2865 were quantified by Taqman qPCR in triplicates using the Roche LightCycler480 system. Primers catpC221-fw, catpC221-rv and catpC221-probe (product length, 132 bp) are listed in Supplementary Table 6. Each reaction mixture (20 µl) contained 10 µl SensiFAST Probe No-ROX Kit 2X (Labgene Scientific), 0.25 µM Probe, 0.9 µM of each primer, and 1 µl (3 ng) of the extracted phage DNA. The standards for $cat_{pC221}$ ranged from $10^7$ copies µl$^{-1}$ in 10-fold dilution to $10^1$ copies µl$^{-1}$, respectively. Initial polymerase activation at 95 °C for 5 min was followed by 45 cycles of denaturation at 95 °C for 10 s, and amplification at 58 °C for 20 s. To exclude the possibility of non-encapsidated DNA contaminants, several controls were added. For this, the absence of non-packaged DNA, DNase I activity, and inactivation were tested as introduced in[37]. Furthermore, the absence of any contaminating extracellular DNA was verified through a no-phage control. These samples were treated equally to regular phage samples, and the absence of extra-cellular plasmid DNA was verified by qPCR after DNaseI treatment. Encapsidation frequencies were calculated as follows: First, detected copy numbers for pUR2865 were normalized to 1 ng of DNA (A). Next, the number of respective phage genome copies in 1 ng DNA (B) was calculated using the following formula[40]: copies = $((1 \text{ ng} × \text{NA} × 10^9) \text{ Mr}^{-1})$, where Mr = size of the phage genomic DNA (bp) multiplied by normalized weight of nucleotide base (650 Da), and NA is the Avogadro constant. Lastly, encapsidation frequencies were calculated using the formula: EF = $(A/(B × C))$, where C indicates the number of plasmids that can be packaged into each respective phage capsid[38] (phage genome (bp) plasmid genome (bp)$^{-1}$).

**Statistics**. All test statistics were calculated with R, using the base package stats. For data manipulation and plotting, tidyverse, dplyr[111], and ggplot2[112] were used. All scripts are available as a rendered R markdown (Goeller_etal21_Rmarkdown. pdf). If not otherwise indicated, an average value is always displayed with its corresponding standard derivation.

**Reporting summary**. Further information on research design is available in the Nature Research Reporting Summary linked to this article.

## Data availability
All isolated bacteriophages are available upon request. The phage genomes sequenced in this study have been deposited as nucleotide sequences in the GenBank database under the accession codes MZ417315-MZ417354 (see Supplementary Data 7, column H) within Bioproject PRJNA663854. The genome of S. epidermidis strain S414 was deposited in the European Nucleotide Archive under the accession code ERZ3124167 (https://www.ebi.ac.uk/ena/browser/view/PRJEB42698). Source data are provided with this paper.

## Code availability
All R scripts used in this study are available as supplemental information in a rendered file (Supplementary Data 10).

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

## Acknowledgements

This work was supported by the Swiss National Science Foundation NRP72 "Antimicrobial Resistance" Project No. 167090, by the European Union's Framework Program for Research and Innovation Horizon 2020 (2014–2020) under the Marie Sklodowska-Curie Grant Agreement No. 659314, and by the ETH Career Seed Grant Project SEED-01 18-1. We kindly thank Carmen Torres, University of La Rioja, Spain; Antonella Demarta, SUPSI-Laboratory of Applied Microbiology, Switzerland; and Vincent Perreten, University of Bern, Switzerland, for providing bacterial strains used in this study. We thank Daniel Fehlmann, WWTP Wädenswil, for providing wastewater samples, determining the chemiophysical parameters, and the stimulating exchange. Further, we would like to thank Hugo Oliveira for sharing python codes and information available for staphylococcal phages in NCBI. We thank Jochen Klumpp and Stefan Handschin for their electron microscopy expert advice. We thank the Functional Genomics Center Zurich (FGCZ) for supporting the library preparation and sequencing. Lastly, we thank Andrea Hauser, Jose Manuel Haro-Moreno, Diana Gutierrez, Jonas Fernbach, and Christian Röhrig for helpful discussions.

## Author contributions

P.C.G. guided and analyzed all experiments and wrote the manuscript. E.G.S. conceived the study, guided experiments, and contributed to the writing of the manuscript. T.E. and P.C.G performed the bioinformatic, network, and encapsidation analysis. T.E. wrote all R scripts. D.L., V.B., N.R., and N.A. contributed to phage isolation. A.N. and P.C.G. propagated sequenced phages. N.R., N.A., and P.C.G. performed the host range assays. E.K. and V.B. established bacterial enrichment cocktails. F.H.C. analyzed sequenced phage genomes phylogenetically and supported bioinformatic analysis. M.J.L. provided conceptual input, partial funding, and corrected the manuscript. All authors read and approved the final manuscript.

## Competing interests

The authors declare no competing interests.
