## [Peer Review File · Nature Communications]

Multi-species host range of staphylococcus phages isolated from wastewaterREVIEWER COMMENTS

Reviewer #1 (Remarks to the Author):

The authors present an extensive characterization of staphylococcal bacteriophages isolated from wastewater from a locality in Switzerland. To my best knowledge, this is the first work to address the host range of phages from wastewater on such a large number of staphylococcal taxa (27 of 32 taxa susceptible to at least one phage), which could be of interest. Determining the network of staphylococcal species with phages is also a valuable result, but the phage families (classes) must be strictly distinguished here. In forty phages of 94 (29 myoviruses and 11 siphoviruses) the authors provide a whole genome sequence. The phages have been classified into two families, only some likely represent new taxa (considering the 50% average nucleotide identity criterion). Using previously described approaches and qPCR, the authors determined the frequency of small plasmid packaging in phages to assess their role in HGT; concerning these results, I doubt if the presence of transducing siphoviruses has been completely eliminated.

Major comments

1) The introduction consists of two separate parts, general questions about the width of host range and summary about staphylococcal viruses. However, lysogeny in staphylococci is omitted. Almost every staphylococcal strain carries prophages, and these prophages (siphoviruses) are mediators of lateral gene transfer, mostly after induction (<https://doi.org/10.1126/science.aat5867>; <https://doi.org/10.1038/ncomms13333>; <https://doi.org/10.1093/femsle/fnw211>). On the other hand, myoviruses are thought to degrade host DNA and are non-transducing due to the mechanism of DNA packaging. The hypotheses that have been addressed should be formulated more precisely, according to the knowledge of the individual phage families.

2) My main criticism is that the authors tar all staphylophages with the same brush when they draw conclusions. Individual phage families, siphoviruses, myoviruses (and podoviruses) differ significantly and share almost no common sequence homologies. While staphylococcal species are relatively conserved in terms of the core genome and, therefore, similar behavior of phage families can be expected in different species. Since you have isolated majority of myoviruses, you cannot conclude that staphylococcal phages have limited specialization and frequently infect multiple species (I suppose it would be true only for one group of phages). What host range do your siphoviruses have? Does the wide range also apply to siphoviruses (only one is marked as Generalist in Table S11)? In Fig. 6, there are presented only 4 siphoviruses with a 40-kb genome for which a narrow host range is generally typical.

On the other hand, myoviruses infecting multiple-species (polyvalent) have been observed previously (usually studied as phages targeting *S. aureus*) (<https://doi.org/10.1006/viro.1998.9203>; <https://doi.org/10.1038/s41598-020-75637-x>).

In this sense, the conclusions should be toned down.

3) Lines 149-152: Please explain which strains were used as propagation strains? How did you excluded spontaneous induction of prophages from propagating strains (myoviruses can mobilize prophages) when preparing stock lysates for phage typing?

4) Broad host range: What was the efficiency of plating (EOP) of tested phages on various species? Since whole genomic sequencing showed high relatedness of several phages, explain, if you isolated really new phages or selected phage mutants or bred the phages on new hosts (see <https://doi.org/10.3390/ph14040325> and <http://dx.doi.org/10.1038/s41598-019-41868-w>). Low EOP values (e.g. few plaques with dilution $10^{>8}$ pfu/ml) may indicate these events.

5) Lines 304-306: Except one case of a giant myovirus, experimental transduction was successful only with siphoviruses. You have not demonstrated gene transfer here. There are mechanisms other than transduction (<https://doi.org/10.1128/mBio.02115-16>).

6) Encapsidation of plasmid DNA: the system described by the authors is applicable only if the propagation strain S414/pUR2865 is prophage-free. The tested phage should be propagated from a single plaque on a prophage-free strain, this is the only way to exclude plasmid encapsulation by resident prophages. Exclusion of prophages by mitomycin induction is not sufficient, the genome of the strain should be sequenced.

Minor comments

1) Line 62: Recent taxonomy status of the Staphylococcaceae is summarized here: <https://doi.org/10.1099/ijsem.0.004498>. But I do not insist on using the genus *Mammaliicoccus* for *S. sciuri* group, it will take a long time to experience.

- 2) Host enrichment cocktails: This chapter is difficult to read, I recommend to summarize the essential facts and move the methodological aspects to M+M section.
- 3) Lines 212-216: In this paper (<https://doi.org/10.1007/s11262-017-1507-2>) the authors described a Kayvirus infecting 13 species.
- 4) Line 309: Why was EM performed only in 57 phages? Phage families can be easily distinguished by EM
- 5) Lines 320: Relatively numerous isolations of siphoviruses with a large genome over 90 kb (7 sequenced genomes) is a significant result. These phages are very rarely isolated as infectious particles and morphological descriptions are missing. On the other hand, these phages are present as defective prophages throughout the Bacillaceae family, usually referred to as SPbeta-like. The comparative genomics of this group deserves a comment in the text.
- 6) Lines 334-337: supplementing the ICTV taxonomy to Fig. 6 and Table 2 (phage genera) would be beneficial
- 7) Lines 371-372 and Fig 7b legend: this is only prediction of genome termini depending on the database used (not detection)

Reviewer #2 (Remarks to the Author):

This work is significant for 1) greatly expanding our knowledge of Staphylococcus phages outside the well-studied *S. aureus* species, with 94 new viruses isolated and 40 new genomes deposited to the public domain; 2) showing that many phages have broad host ranges across the species, and can presumably act as conduits for Horizontal gene transfer. This work represents a major research effort and the methods are well-reported.

The major downside is that the genomes of the host strains are not available. This means that the authors are forced to speculate about the reasons for individual phage host range rather than being able to propose mechanisms based on genome analysis. Phages that can cross species barriers may recognize similar WTA structures.

The discussion should address whether the enrichment method (using a cocktail of multiple species) potentially enrich for phages with cross-species host ranges. Also, another open question regarding the global significance is whether this collection of phage host ranges are likely representative of Staphylococcus phages in general. Is it possible that phages sampled only from one geographic location can be thought to represent the whole genus?

Specific Points

Line 45 "On average, they infect two strains from a single species". I suggest deleting this sentence as it refers to a particular study and the results cannot be generalized across all phages.

Line 140 – what does a "50% redundancy in phage isolation mean"?

Line 230-242. Significance of this section is unclear. Is there a plausible reason why antibiotic-resistant strain would be more or less susceptible to phages?

Line 399-402. Without any knowledge about the WTA structure of the strains in the collection (a weakness of this study), this is a premature statement. It could be possible that there is a diversity of WTA structures within Staph species (this is known in *S. epi* and *S. aureus*) and phages recognize common structures in different species.

Line 421, "Second, the .." this sentence needs to be rewritten for clarity.

Line 512. There needs to be a supplementary table listing the isolation date and isolation location of the 132 strains in the work.

Line 517 – Supplementary Table 10 does not give this information

Line 583 – not clear what is meant by "BioPython 32"

Fig 1 – why are Aureus strains dark blue, if this species is a CoPS?

I suggest that all RMarkdown scripts used in this analysis be made available as supplemental information or through a repository such as github.

Reviewer #3 (Remarks to the Author):

The manuscript has described that the host range of phages among *Staphylococcus* spp. I found several major issues of this work.

1. I understand the authors have investigated the host range of the isolated phages. The aim of the study is not clear. Phage therapy or HGT?
2. The authors isolated only two of three major types of *Staphylococcus* phages. Myophages with high transduction activity are not included in the study. Probably, different type of phages. It is a pity not to include podovirus in the study. The authors may describe the isolated phages based on ICTV virus taxonomy.
3. The authors claimed the potential HGT via phages. However, the work has not provided sufficient amounts of HGT experiments for publication in my opinion. The authors should HGT experiment among *Staphylococcus* spp, using different phage types in my opinion.

Minor comments

1. The authors should samples phages from different sources. Since sewage system is different among regions and countries, the authors should describe the sewage system that the authors approached.
2. The authors just sampled phages from sewage water using different *Staphylococcus* spp. In my opinion, the authors should screen the phages different sources (e.g., animal farms).

We would like to thank the reviewers for the insightful comments. We addressed all the issues and questions raised in this revision and hope that you will find this new version of the manuscript suitable for publication.

Sincerely,

the authors

Point by point responses to reviewers' comments

Reviewer 1

The authors present an extensive characterization of staphylococcal bacteriophages isolated from wastewater from a locality in Switzerland. To my best knowledge, this is the first work to address the host range of phages from wastewater on such a large number of staphylococcal taxa (27 of 32 taxa susceptible to at least one phage), which could be of interest. Determining the network of staphylococcal species with phages is also a valuable result, but the phage families (classes) must be strictly distinguished here. In forty phages of 94 (29 myoviruses and 11 siphoviruses) the authors provide a whole genome sequence. The phages have been classified into two families, only some likely represent new taxa (considering the 50% average nucleotide identity criterion). Using previously described approaches and qPCR, the authors determined the frequency of small plasmid packaging in phages to assess their role in HGT; concerning these results, I doubt if the presence of transducing siphoviruses has been completely eliminated.

Major comments

1) *The introduction consists of two separate parts, general questions about the width of host range and summary about staphylococcal viruses. However, lysogeny in staphylococci is omitted. Almost every staphylococcal strain carries prophages, and these prophages (siphoviruses) are mediators of lateral gene transfer, mostly after induction* (<https://doi.org/10.1126/science.aat5867>; <https://doi.org/10.1038/ncomms13333>; <https://doi.org/10.1093/femsle/fnw211>). *On the other hand, myoviruses are thought to degrade host DNA and are non-*

transducing due to the mechanism of DNA packaging. The hypotheses that have been addressed should be formulated more precisely, according to the knowledge of the individual phage families.

We appreciate the reviewer request and have re-written part of the introduction to focus more on the distinct phage families and their respective attributes (lines 53-72). Yet, we cannot go into too much detail about the different mechanisms of transduction, as our focus is on the plaquing phage host range pattern, and the possibilities which a generalized broad host range opens for staphylococcal phages (e.g., the connection of different biomes or species). In future work, we aim to assess how wide and well the detected transducing particles can transduce material across the here established network. This will also include more detailed investigations on the precise mechanisms of transduction.

2) My main criticism is that the authors tar all staphylophages with the same brush when they draw conclusions. Individual phage families, siphoviruses, myoviruses (and podoviruses) differ significantly and share almost no common sequence homologies. While staphylococcal species are relatively conserved in terms of the core genome and, therefore, similar behavior of phage families can be expected in different species. Since you have isolated majority of myoviruses, you cannot conclude that staphylococcal phages have limited specialization and frequently infect multiple species (I suppose it would be true only for one group of phages).

In total, we have isolated 94 different staphylococcal phages from wastewater. Unfortunately, we only know the morphology of roughly half of them. For those, the reviewer is correct that these were predominantly myoviruses, i.e., 44 myoviruses and 12 siphoviruses (of which five belong to Cluster B (induced from the bacterial fraction), and 7 to cluster D, according to classification by Oliveira et al. <https://www.ncbi.nlm.nih.gov/pmc/articles/PMC6507118/>). Of all 94 phages, only 4 infect a single species of which 3 are cluster B phages (induced) and one cluster D siphovirus. If we look at just the siphovirus fraction, we note that 4 out of 12 viruses are species specific, whereas the rest can replicate on more than one species. An important feature of cluster B siphoviruses to point out is that they can integrate into the bacterial genome, and therefore, assessing the productive infection host range may not reflect the true underlying host range. Considering the remaining fraction for

which we do not know the morphology, all phages infected at least two distinct species. In this fraction, we assume to not only have recovered myoviruses.

It is true that myoviruses seem to have a broader host range compared to the siphoviruses with 90 kb or 40 kb genomes, as described in lines 335-343 and displayed in Figure 6. One could propose that the larger the phage genome, the more space for proteins that counteract host defences, thus, a larger host range (<https://doi.org/10.1111/ele.13630>). On the other hand, we also find it difficult to compare the plaquing host ranges of temperate phages and virulent phages. These arguments, together with the unknown morphology of the subset of ~40 phages, and the presence of only 4 species-specific phages, led us to the motion that generally, staphylococcal phages might not be as species-specific as we thought. As a further remark, the goal of this study was to point out the complex and wide range of interactions of bacterial populations when a certain phage population is present in an environmental niche. Therefore, we find it interesting to partially address all phage host ranges as a whole network, rather than focusing on individual phage families. To this point, all here isolated phages were recovered from a single niche in a single a sampling effort.

What host range do your siphoviruses have? Does the wide range also apply to siphoviruses (only one is marked as Generalist in Table S11)?

The isolated siphoviruses generally feature a narrower host range compared to myoviruses (both in numbers of infected strains and species). Nevertheless, 8 out of 12 phages infect more than one species. The term generalist was used here not to describe phages with broad or narrow host ranges, but rather to characterize the proportions of strains belonging to a respective species within the infected bacteria. For example, even though phage K is a broad host range phage, we characterize it as a specialist, as it predominantly infected *S. aureus* strains (*S. aureus* strains comprised > 50% of all infected strains for this phage). As another example, if a siphovirus infected four *S. epidermidis* strains, and one *S. sciuri* strain, we consider this phage a specialist, because *S. epidermidis* strains make up most of the infected host strains. Nevertheless, this phage still replicates on more than one species (and could therefore be considered as a broad host range phage).

In Fig. 6, there are presented only 4 siphoviruses with a 40-kb genome for which a narrow host range is generally typical. On the other hand, myoviruses infecting multiple-species (polyvalent) have been observed previously (usually studied as phages targeting S. aureus) (<https://doi.org/10.1006/viro.1998.9203>; <https://doi.org/10.1038/s41598-020-75637-x>).

In this sense, the conclusions should be toned down. To properly incorporate the discussion points above, we toned down our conclusions (line 401, 421) and improved discrimination between the host range of siphoviruses versus myoviruses (lines 337-343).

3) Lines 149-152: Please explain which strains were used as propagation strains? How did you excluded spontaneous induction of prophages from propagating strains (myoviruses can mobilize prophages) when preparing stock lysates for phage typing?

For preparation of the stock lysate used for host range spotting, we amplified all phages on their respective isolation host (27 strains from 15 different species). Unfortunately, we cannot exclude a spontaneous induction of prophages, as most staphylococcal strains (as pointed out) seem to harbour internal prophages. We acknowledge that propagation on well characterized lab-strains or prophage-cured propagation hosts would be the best solution. However, one would also limit the variety of phages to include into host range testing, since most of them do not plaque on lab strains (e.g., none of the phages infected *S. aureus* RN4220). We assume that any putative induced prophage would be present at much lower numbers compared to the phages used in propagation, as induction usually yields titres around 10^3 - 10^4 pfu/ml (unpublished observation). If one uses phage concentrations from 10^4 - 10^8 for spot-assay host range testing any induced background phage would most likely only be visible in the higher concentration spots (please also see the next response below). Furthermore, when we analysed phage concentrates using EM, we could not detect any contaminating phage with siphoviral morphology in the myovirus phage samples. In summary, we acknowledge the fact that there are some limitations to the accuracy of the resultant host range assay, e.g. the influence of potential contaminating lysogenic phages, or the induction of lysogens from the bacterial lawn on which we spot the phage concentrates (additional detailed explanation can be found in the next comment).

4) *Broad host range: What was the efficiency of plating (EOP) of tested phages on various species? Since whole genomic sequencing showed high relatedness of several phages, explain, if you isolated really new phages or selected phage mutants or bred the phages on new hosts (see <https://doi.org/10.3390/ph14040325> and <http://dx.doi.org/10.1038/s41598-019-41868-w>). Low EOP values (e.g. few plaques with dilution 10^8 pfu/ml) may indicate these events.*

After their initial isolation (three rounds of individual plaque picking), phages were amplified once on the respective isolation host, and the same phage lysate was used for all spots on the 123 hosts. Regarding host range determination, we performed assays using phage dilutions from 10^8 pfu/ml down to 10^4 pfu/ml. Generally, we counted all interactions as positive when single spots appeared in one of the dilutions. However, in most cases, we saw a decreasing effect from complete lysis to individual plaques only in the two lowest dilutions. Examples are shown in the two following images.

In host M1997-2/10, all spotted phages readily infected this strain (two phages per row, five dilutions per phage, spots left to right from 10^8 pfu/ml to 10^4 pfu/ml). For C6866 (right scan), positive interactions were only recorded for the phages not enclosed in a red box. For phages in a red box, a lysis from without event was reported.

Technically, the generation of phage mutants might be feasible as we isolated all phages for three sequential rounds on their isolation hosts for phage stock preparation. During these rounds of isolation, it is possible, that the isolated phage

mutated to better adapt to the respective host. This would mean that one phage, isolated on different hosts, would give rise to two almost identical phages with maybe slightly different host range patterns. Thus, we cannot exclude a potential breeding effect. However, it is also possible that almost identical phages give rise to different host range patterns without the event of phage breeding. One exemplary scenario is, that an identical phage was isolated and amplified on two different hosts, and thus features two slightly different host ranges, as host methylation patterns of the respective amplification hosts could determine the activation or inaction of internal defence mechanisms regarding the incoming phage on the next host. Thus, host ranges can change depending on which host the phage replicated last. This would explain why some phages had different host ranges, but their genomes differ in only some point mutations. Moreover, we also have to acknowledge that few point mutations could give rise to a different host range (also independent of breeding), as point mutations can help the phage to evade CRISPR/cas detection or restriction modification systems, or can alter the receptor binding domains. As a last remark, we want to highlight, that through phage isolation from one specific niche at one specific timepoint, we already obtain a snapshot of phage evolution within the respective sampling spot. Phages are known to mutate at very high frequencies, and we cannot exclude that two highly similar phages already existed as such in the initial phage isolation sample.

Regarding the efficiency of plating, we actually analysed the EOP for all "outlet" phages on each respective strain and species (data not shown in the manuscript). As an example, the graph below shows the data visualized as the relative EOP compared to the best infected host for phage PG-2021_47.

Relative Efficiency of Plating for $\phi 47$ on different *Staphylococcus* hosts

Interestingly, strains from the same species do not always cluster together, and the EOP may vary significantly between strains of the same species. Therefore, the EOP is not necessarily driven by species, but more likely by the characteristics of each respective strain, as proposed by Bernheim and Sorek (<https://doi.org/10.1038/s41579-019-0278-2>). The authors suggest that it is more likely that individual strain characteristics, such as internal defence mechanisms, determine how well the phage can infect a specific strain. The species barrier (frequently represented by bacterial surface receptors) may be highest, but not the exclusive determinant of a phage host range. Therefore, receptor availability on the bacterial species determines if a phage can infect or not, but the infection outcome and efficiency is highly variable due to diverse internal defence mechanisms. Furthermore, WTAs, the proposed receptors for staphylococcal phages, can be identical within two different species, but can also vary between the same species (as known for *S. epidermidis* and *S. aureus*). Therefore, we propose that the EOP is more influenced by internal mechanisms, rather than the species “barrier” for staphylococcal phages.

Regarding the appearance of individual plaques using for example the 10^8 dilution, there may be potential biases regarding the host range assay. First, the lysate could harbour an induced prophage (see answer above) which will extend the true host range of the tested phage. We believe, however, that induced prophages would be

present at much lower numbers compared to the amplified stock phage, and therefore, can only affect the highest phage concentrates. Next, we need to consider that the strain on which we spot the phage concentrate might harbour a prophage, which could also be induced and give rise to plaques that bias the results. Nevertheless, the number of possible plaques is limited by the dilution, and as long as one sees the expected dilutional effect from 10^8 to 10^4 pfu/ml, it should be the result of the high concentrated stock phage in the lysate. The reason why we chose to spot this phage collection from 10^8 pfu/ml to 10^4 was that we wanted to see individual plaques on the lower dilutions (which most often only occurred from 10^5 - 10^4 pfu/ml) to rule out potential “lysis from without” effects, which would impede discrimination from productive infections.

5) *Lines 304-306: Except one case of a giant myovirus, experimental transduction was successful only with siphoviruse. You have not demonstrated gene transfer here. There are mechanisms other than transduction (<https://doi.org/10.1128/mBio.02115-16>).*

This is correct. The sentence was adapted to: “Our findings unveil phages to connect copious hosts from multiple species, ecosystems, and of different clinical relevance, and place phages as potential vectors for bacterial genetic exchange.” Lines 303-305.

6) *Encapsidation of plasmid DNA: the system described by the authors is applicable only if the propagation strain S414/pUR2865 is prophage-free. The tested phage should be propagated from a single plaque on a prophage-free strain, this is the only way to exclude plasmid encapsulation by resident prophages. Exclusion of prophages by mitomycin induction is not sufficient, the genome of the strain should be sequenced.*

A valid point. We have sequenced this strain using Illumina and could not detect any putative prophage. Details can be found in the M+M section (lines 638-643) and in the results part (line 358-359).

Minor comments

1) *Line 62: Recent taxonomy status of the Staphylococcaceae is summarized here:*

<https://doi.org/10.1099/ijsem.0.004498>. But I do not insist on using the genus *Mammaliococcus* for *S. sciuri* group, it will take a long time to experience.

Thank you for this remark. As some of the *Staphylococcus* species used here were classified into the new genus *Mammaliococcus*, and some subspecies were reclassified into further individual species, we added an “Addendum in Revision” section, where we denoted those recent changes in taxonomy. This new section can be found in line 687-692.

2) *Host enrichment cocktails: This chapter is difficult to read, I recommend to summarize the essential facts and move the methodological aspects to M+M section.*

We agree and have simplified this chapter and moved sections to the M+M, lines 489-495.

3) *Lines 212-216: In this paper (<https://doi.org/10.1007/s11262-017-1507-2>) the authors described a Kayvirus infecting 13 species.*

We feel that the host range of the kayviruses has not been properly characterized. This is due to the limited number of strains and species one could test in culture-dependent infection assays. We look forward to new insights regarding the extend of phage host range, as new techniques are introduced and established.

4) *Line 309: Why was EM performed only in 57 phages? Phage families can be easily distinguished by EM*

Initially, we selected 56 phages for whole genome sequencing and performed EM for all of them. The morphology of phage K was already described (thus 57 phages). However, some of the phages failed to produce a sufficiently high titre for DNA extraction, and others gave troubles in library preparation. This is why we ended up with a reduced number of 40 sequenced phages, compared to 57 phages with EM. It is true that EM can be performed relatively easy and may be useful to distinguish diverse phage families. However, for high quality results, time-consuming and labour-intensive caesium chloride purification is required. This posed an immense workload and restricted the number of phages we could visualize using EM. We have added this information in the M+M section “electron microscopy”, line 557-587.

5) *Lines 320: Relatively numerous isolations of siphoviruses with a large genome over 90 kb (7 sequenced genomes) is a significant result. These phages are very rarely isolated as infectious particles and morphological descriptions are missing. On*

the other hand, these phages are present as defective prophages throughout the Bacillaceae family, usually referred to as SPbeta-like. The comparative genomics of this group deserves a comment in the text.

Thank you for this suggestion. According to Oliveira *et al* (PMID: 31072320), the group of 90 kb siphoviruses (Cluster D) is a group of virulent siphoviruses with, however, only three representatives so far. They are described with genome sizes between 89-93 kb and do not possess any predicted lysogeny control functions. Oliveira *et al* highlight and describe the morphological characteristics (long tails > 300 nm), as well as the genome organisation of this group of phages. In contrast to the Cluster D phages, the SP-beta-like phage is a singleton within the staphylococcal phages (also describe by Oliveira *et al*) and has an even larger genome size of >120 kb. Of the 11 sequenced siphoviruses in our study, 7 feature a genome size ~90 kb, the morphological characteristics of very long tails, and lack of a lysogeny module. Accordingly, they belong to cluster D siphoviruses. In Figure 6 within the *Siphoviridae* (and see below), one can further see that the 90 kb viruses isolated here are much closer related to members of the Sextaevirus genus (cluster D phages) as to the SP-beta-like singleton. We state these findings in lines 315, 319-321 and describe the 90 kb phages as members of cluster D siphoviruses. Nevertheless, we agree that these phages greatly extend the previously narrow group of 90 kb viruses.

6) Lines 334-337: supplementing the ICTV taxonomy to Fig. 6 and Table 2 (phage genera) would be beneficial

We agree. For Figure 6, we now include the phage genera for each of the representative phage per genus (this information was taken from the previously established NCBI classification). Personal communication with an ICTV member, led to the conclusion that it would be difficult to map a new phage genome to an existing genus, or even to establish a new genus within the existing families. If properly done, such a process would require a significant bioinformatic computation and the output is usually highly dependent on the algorithm used. A further hurdle in the classification of new phages as of now is that ICTV is currently changing from morphological-based classification to genome-based classification. In addition, there is no uniform scheme (and tools) for the assignment of phages to genera yet. Therefore, we decided to not classify the phages to a higher level than the families that were established and confirmed by the ICTV.

7) Lines 371-372 and Fig 7b legend: this is only prediction of genome termini depending on the database used (not detection)

Thank you for pointing this out. The sentence was changed to: "Thus, we predicted the phage genome termini *in silico*, which should reflect the phages' DNA packaging mechanisms." (line 378). In the figure legend, we changed the wording from detection to prediction (line 828).

Reviewer 2

This work is significant for 1) greatly expanding our knowledge of Staphylococcus phages outside the well-studied S. aureus species, with 94 new viruses isolated and 40 new genomes deposited to the public domain; 2) showing that many phages have broad host ranges across the species, and can presumably act as conduits for Horizontal gene transfer. This work represents a major research effort and the methods are well-reported.

The major downside is that the genomes of the host strains are not available. This means that the authors are forced to speculate about the reasons for individual phage host range rather than being able to propose mechanisms based on genome analysis. Phages that can cross species barriers may recognize similar WTA structures.

The discussion should address whether the enrichment method (using a cocktail of multiple species) potentially enrich for phages with cross-species host ranges.

We thank the reviewer for this comment. First, we would like to point out that phage enrichment and isolation protocols exist that selectively target the broad host range phages by passaging phage suspensions sequentially on multiple hosts, and only recover those that underwent successful amplification on all selected hosts (<https://doi.org/10.1128/aem.02382-15>). In contrast to such procedures, we aimed to isolate a most diverse community of phages by either amplifying them non-sequentially on a broad set of hosts, or by induction of prophages. The reason to include as many enrichment hosts as possible was only to gain a maximally diverse community of phages, rather than selectively harvesting broad host range phages. However, we acknowledge that by incorporating many enrichment hosts, one could assume that we also selectively enrich broad host range phages, as those phages are most likely to amplify on more than one host and therefore might overgrow with respect to more selectively replicating phages. The scenario that broad host range phages are multiple times isolated on diverse species or strains and are therefore preferentially, is plausible. Interestingly, we observed the contrary, i.e., the broadest host-range phage (as for example phage PG-2021_17 infecting 18 species), was only isolated once and phages were rarely isolated on multiple species. Generally, only 10 phages were isolated on more than one species (details on the isolation frequency can be found in supplementary table 9), whilst 90 phages infect more than one species, but were isolated on only a single one. For clarity, we made changes in lines 145 and lines 148-151. The most frequently isolated phage (PG-2021-5) was isolated 16 times and is a 90 kb siphovirus that infects “only” 10 strains from 5 species. Overall, of the phages with known morphology, 6 out of 7 of the 90 kb phages were isolated more than once, whereas only 9 out of 44 myoviruses were isolated multiple times. Interestingly, the myoviruses displayed a broader host range compared to the 90 kb siphoviruses. Ultimately, those observations remain to be quantified, as we do not know the morphology of all isolated phages. However, these observations led us to the hypothesis of a possible trade-off between infecting many diverse strains/species and the phage fitness on each of the respective hosts. In other words, broad host range phages might have the opportunity to amplify on multiple strains, but cannot outgrow the more selective phages, as they have an

increased fitness (and thus better amplification rates) on their respective host. Of course, this scenario is only valid for phages replicating through the lytic cycle (which most likely represent the majority of our collection). Overall, we believe to have isolated the most abundant phages after three successive enrichment rounds, which are either phages that were previously most abundant in the native viral fraction, or the phages that could best amplify on one or more of the host bacteria.

Also, another open question regarding the global significance is whether this collection of phage host ranges are likely representative of Staphylococcus phages in general. Is it possible that phages sampled only from one geographic location can be thought to represent the whole genus?

We agree that the host range collection determined here may not be representative for the whole genus *Staphylococcus*. Some phage morphologies are not represented (such as podoviruses), and new types of viruses are likely to be discovered for the staphylococci (as for example the giant myoviruses). However, the host range profiles provided here are the most comprehensive one established so far, and is based upon diverse isolation approaches to include both the profile of free viral particles as well as of those integrated into the bacterial chromosomes. Based on these 94 phages, we can presume that the host range trend for staphylococcal phages is less species specific as previously assumed. This highlights the possibilities for phages to act as vectors for HGT between diverse representatives of the genus *Staphylococcus*. Regarding the geographic location, it can be safely assumed that phages from diverse environments may act differently regarding host ranges (we added this information in lines 410-412). The host range for phages in diverse environments might be dependent on the host diversity available in each specific niche. Nevertheless, we believe that phages found in wastewater actually represent a community that originated from diverse ecosystems. Therefore, if an ecosystem must be chosen to represent a most diverse variety of staphylococcal (or other) phages, it may likely be wastewater.

Specific Points

Line 45 "On average, they infect two strains from a single species". I suggest deleting

this sentence as it refers to a particular study and the results cannot be generalized across all phages.

We agree. The second part of this sentence was deleted (line 37), and the Introduction rewritten to improve clarity.

Line 140 – what does a “50% redundancy in phage isolation mean”?

Initially, we isolated 179 phages during the three isolation rounds, on different enrichment/isolation hosts, and based onto differences in plaque morphologies. However, it was not always the case that different plaque morphologies indicate different phages. To discriminate how many unique phages we have isolated from the 179 collected, we determined their host range on 123 strains. All phages with an identical host range were clustered in groups, and we continued working with only one phage per cluster. With this analysis, we went from 179 phages down to 94 unique ones, which indicates almost 50 % redundancy in phage isolation. It is important to note that this does not imply that close to half of the phages were isolated twice. Over 60 % of the phages were isolated only once (they displayed a unique host range), whereas we found to have isolated 32 phages at multiple times. At highest, we have isolated a phage (PG-2021_5) 16 times on 7 different strains from three species (probably because it was efficient in amplification on many different enrichment hosts).

For clarity, this part of the manuscript was rephrased (lines 134-136, line 141).

Line 230-242. Significance of this section is unclear. Is there a plausible reason why antibiotic-resistant strain would be more or less susceptible to phages?

We agree that this section was not totally clear and have rewritten this part (lines 236-246). We now highlight that our goal was to assess whether phages infect strains from different ecosystems to analyse if they have the potential to act as shuttle vectors for genetic material across biomes. Furthermore, for phages to play a role in the dissemination of antibiotic resistance, the ultimate pre-requisite would be to infect drug-resistant strains on which the phage progeny might capture an antibiotic resistance gene.

Line 399-402. Without any knowledge about the WTA structure of the strains in the collection (a weakness of this study), this is a premature statement. It could be possible that there is a diversity of WTA structures within Staph species (this is known in S. epi and S. aureus) and phages recognize common structures in different species.

We agree with the reviewer here that this is hypothetical and modified this sentence to:

“we hypothesize that WTAs structures might not be that evolutionary divergent⁵³; phages might recognize conserved structures on different WTAs, or phages might recognize alternative, more conserved receptors on the bacterial cell wall⁵².” Lines 407-412.

⁵³ This paper indicates that some staph phages may bind to the peptidoglycan and that WTAs as not as important for binding as we think: <https://doi.org/10.1038/srep4631>.

⁵⁴ This paper points that *S. aureus* and *S. epidermidis* could express similar WTAs due to a horizontally transferred gene cluster: <https://doi.org/10.1038/s41564-021-00913-z>.

Line 421, “Second, the ..” this sentence needs to be rewritten for clarity.

Thank you for pointing this out. This line was rewritten to: “The second bottleneck applies to the delivery and expression of the cargo DNA in the cytoplasm of the recipient bacteria” (lines 428-429).

Line 512. There needs to be a supplementary table listing the isolation date and isolation location of the 132 strains in the work.

We apologise because the supplementary information (Supplementary Table 10) linked to this paragraph was numbered incorrectly. It should be supplementary Table 8 (line 532). Therefore, the information about the origins of the strains was missing. In supplementary Table 8, all known characteristics of the 123 strains used in this study are included. We indicate, for each strain, the reference paper and the date of publication (column O), the isolation country (and city if known) in column E, and the

site of sampling (e.g. which site on the human body or animals, or if it was a clinical or asymptomatic infection, or from which environment the strain originated, e.g. WWTP, surface water etc.). We have also tried to track the approximate isolation dates for the strains and their exact location; however, this information is not always provided in the publication. This is probably due to clinical studies that lasted over several years, and from which most strains were taken. Therefore, in many cases and when strains originated from large culture collections from hospitals or academia, there is no isolation date available, only the publication date.

Line 517 – Supplementary Table 10 does not give this information

We apologize for this mishap. Information was changed to “Supplementary table 8” (line 532).

Line 583 – not clear what is meant by “BioPython 32”

We changed this information to: “Bio.Entrez package from Biopython within a conda environment was used to retrieve fully sequenced staphylococcal phage genomes deposited at GenBank as of June 2020 (n = 292).” Lines 600-601.

We hope this clarifies the situation. The initial wording was taken from Oliveira *et al* (<https://www.ncbi.nlm.nih.gov/pmc/articles/PMC6507118/>) who provided us with the code for this specific analysis.

Fig 1 – why are Aureus strains dark blue, if this species is a CoPS?

The cluster group, which includes all strains from *S. epidermidis* and *S. aureus*, as well as 7 more species in this study (*S. capitis*, *S. caprae*, *S. warneri*, *S. pasteurii*, *S. haemolyticus*, *S. hominis*, *S. lugdunensis*), is called “Epidermidis-Aureus” as established in <https://doi.org/10.1186/1471-2148-12-171>.

In the referred graph, the strains are sorted according to the indicated publication from *S. aureus* to *S. epidermidis* and then the other 7 species, from left to right. Therefore, *S. aureus* strains (very few infections) are in light blue, *S. epidermidis* strains plus the other CoNS species are in dark blue. We believe this clarifies this situation.

I suggest that all RMarkdown scripts used in this analysis be made available as supplemental information or through a repository such as github.

Thank you for the suggestion. We have done this for the revision and have encoded this statement in the code availability section:

“All R scripts used in this study are available as supplementary information in rendered files (.pdf)”. Lines 684-686.

Reviewer 3

The manuscript has described that the host range of phages among Staphylococcus spp. I found several major issues of this work.

1. I understand the authors have investigated the host range of the isolated phages. The aim of the study is not clear. Phage therapy or HGT?

We thank the reviewer for pointing out the lack of clarity. Our main goal was to show the potential role of staphylococcal phages in HGT, especially in the spread of antimicrobial resistance determinants. But we agree that in the introduction, this aim may have been neglected a bit. We have therefore rewritten the introduction to clarify the main research goal, and also adapted it to the comments from reviewer 1 (e.g., to focus more on the characteristics of each phage family within the genus staphylococcus). Changes are in highlighted in blue.

2. The authors isolated only two of three major types of Staphylococcus phages. Myophages with high transduction activity are not included in the study. Probably, different type of phages. It is a pity not to include podovirus in the study. The authors may describe the isolated phages based on ICTV virus taxonomy.

Thanks for this comment. It is true that we only characterized phages with three distinct morphologies, e.g., small and large siphoviruses, and myoviruses. We also think it is unfortunate we did not have any podoviruses in the fraction we have visualized with EM. A possible explanation for the absence of podoviruses in the characterized phage fraction (56/94) is their generally very narrow host range. Furthermore, podoviruses for staphylococcal phages are generally rarely isolated, as only 16 of 205 (June 2018, doi: 10.1186/s12864-019-5647-8) described staphylococcal phages have podoviral morphology. In addition, we think it was rather unlikely to have isolated a large myovirus with high transduction activities (e.g. phage S6), as this type of viruses was only isolated once so far for staphylococci (<https://doi.org/10.1038/ismej.2014.29>).

For the ICTV virus taxonomy description, we have included the known genera for each respective phage family within the phylogenomic tree (Figure 6). This missing classification was also pointed out by reviewer 1, comment 6 in minor comments. Please refer to detail there.

3. The authors claimed the potential HGT via phages. However, the work has not provided sufficient amounts of HGT experiments for publication in my opinion. The authors should HGT experiment among Staphylococcus spp, using different phage types in my opinion.

Reviewer points out an apparent lack of HGT data. However, the main goal of this study was not to demonstrate HGT for individual phages, but rather characterize the extent to which phages could serve as vector agents for gene traffic when they are present as natural communities among different bacterial species. We claim that phages could be potential agents of HGT because we observe and describe a dense interspecies network among the genus *Staphylococcus*, apparently mediated through phages. However, we acknowledge that for phages to govern HGT, each individual phage must be assessed for their ability to do so. Regarding this, the first bottleneck is the infection of diverse species (which we show), and the ability to incorporate foreign genetic material from the donor bacterium (which we demonstrate using a small resistance plasmid). The next main bottleneck to address would be the delivery and expression of a DNA cargo (the small plasmid) by those phages for which we could show high incorporation frequency of non-viral genetic material. Furthermore, one could set up experiments to show how far genetic material could travel across the established network once a suitable phage and DNA cargo is identified. We intend to address the last two aspects in upcoming studies, in which we will focus more on HGT and the three distinct modes of transduction, rather than on the breadth of the phage host range and encapsidation frequencies. The main point of this manuscript is the broad host range of staphylococcal phages and the dense network between staphylococcal species governed by phages.

Minor comments

1. The authors should sample phages from different sources. Since sewage systems are different among regions and countries, the authors should describe the sewage system that they approached.

There is no question that sewage systems differ between regions and countries. We extended the description of the sewage system we sampled in the Material and Method section (lines 444-447). We agree sampling different sources would increase the possibilities to find additional novelties and the results may differ from those we present. We selected a WWTP because it is considered a hotspot for microbial diversity and assembly from different sources, including industry, households and clinical settings. We consider this an optimal source to address our research question.

*2. The authors just sampled phages from sewage water using different *Staphylococcus* spp. In my opinion, the authors should screen the phages from different sources (e.g., animal farms).*

We focused on phage isolation from sewage, as WWTPs are considered a hotspot for bacterial recombination and HGT, due to the very high microbial density and external stressors such as residual antibiotics or inducing chemicals (<https://doi.org/10.1016/j.tim.2020.12.011>; <https://doi.org/10.3389/fmicb.2017.02298>). Since we were most interested in (1) the connection potential of natural phages to diverse bacteria, and (2) the possible role of phages in the spread of antimicrobial resistance, we sampled phages from wastewater only. However, we agree that phages present in other ecosystems may behave differently from what we describe here. It should also be noted that phages from wastewater do not necessarily originate from the wastewater pool, but actually stem from diverse environments that eventually drain into the treatment plant. Therefore, one could consider those phages to originate from extremely diverse sources. In future studies, it would be interesting to include phages sampled from other hotspots, such as animal farms or manure.

REVIEWERS' COMMENTS

Reviewer #1 (Remarks to the Author):

The manuscript by Göller and co-authors was significantly improved and much easier to read and follow the logistic of the results. In the rebuttal letter, the authors also explained some of my critical comments. I am satisfied with how the authors addressed my original issues. I also feel that the authors responded properly to the second and the third reviewer's comments.

Reviewer #2 (Remarks to the Author):

I have read through the response to all reviewer comments. I believe the authors have thoughtfully analyzed each comment and in most cases made appropriate modifications to the manuscript and supplement. This was already a strong study pre-review and it has been improved further through the reviewing process.